# Differences in ozone retrieval in MIPAS channels A and AB: a spectroscopic issue

Norbert Glatthor[1], Thomas von Clarmann[1], Gabriele P. Stiller[1], Michael Kiefer[1], Alexandra Laeng[1], Bianca M. Dinelli[2], Gerald Wetzel[1], and Johannes Orphal[1]

[1]Karlsruhe Institute of Technology, Institute of Meteorology and Climate Research,Karlsruhe, Germany
[2]Institute of Atmospheric Sciences and Climate (CNR-ISAC), Bologna, Italy

**Correspondence:** N. Glatthor (norbert.glatthor@kit.edu)

**Abstract.** Discrepancies in ozone retrievals in MIPAS channels A ($685$–$970$ cm$^{-1}$) and AB ($1020$–$1170$ cm$^{-1}$) have been a long-standing problem in MIPAS data analysis, amounting to an inter-channel bias (AB-A) of up to $8\%$ between ozone volume mixing ratios in the altitude range $30$–$40$ km. We discuss various candidate explanations, among them forward model and retrieval algorithm errors, inter-channel calibration inconsistencies, and spectroscopic data inconsistencies. We show that forward modelling errors as well as errors in the retrieval algorithm can be ruled out as an explanation because the bias can be reproduced with an entirely independent retrieval algorithm (GEOFIT) relying on a different forward radiative transfer model. Instrumental and calibration issues can also be refuted as explanation because ozone retrievals based on balloon-borne measurements with a different instrument (MIPAS-B) and an independent level-1 data processing scheme produce a rather similar inter-channel bias. Thus, spectroscopic inconsistencies in the MIPAS database used for ozone retrieval are practically the only reason left. To further investigate this issue, we performed retrievals using additional spectroscopic databases. Various versions of the HITRAN database generally produced rather similar channel AB-A differences. Use of a different database, namely GEISA-2015, led to similar results in channel AB, but to even higher ozone volume mixing ratios for channel A retrievals, i.e. to a reversal of the bias. We show that the differences in MIPAS channel A retrievals result from about $13\%$ lower air-broadening coefficients of the strongest lines in the GEISA-2015 database. Since the errors in line intensity of the major lines used in MIPAS channels A and AB are reported to be considerably lower than the observed bias, we posit that a major part of the channel AB-A differences can be attributed to inconsistent air-broadening coefficients as well. To corroborate this assumption we show some clearly inconsistent air-broadening coefficients in the HITRAN-2008 database. The inter-channel bias in retrieved ozone amounts can be reduced by, e.g., increasing the air-broadening coefficients of the lines in MIPAS channel AB in the HITRAN-2008 database by $6$–$8\%$.

## 1 Introduction

Ozone is one of the most important trace gases in the atmosphere. Stratospheric ozone to a large extent prevents solar ultraviolet (UV) radiation from reaching the Earth's surface. On the other hand tropospheric ozone is a harmful air pollutant. Therefore knowledge of its atmospheric concentration is of high interest. Remote sensing of ozone is performed in a wide spectral range

covering the microwave, infrared and ultraviolet regions. For measurements in the infrared the strong ozone $\nu_3$ band around 10 $\mu$m is of particular interest. To obtain accurate atmospheric ozone volume mixing ratios (VMRs) high quality line parameters are required. For this reason a large number of laboratory measurements has been performed. On the basis of three independent laboratory studies Flaud et al. (2003a) compiled a dedicated line list for evaluation of MIPAS ozone measurements, which

has been included in the HIgh-resolution TRANsmission database version 2004 (HITRAN-2004) (Rothman et al., 2005) and later ones. A review of various laboratory studies performed during the past decades to determine ozone line intensities in the 9–11 $\mu$m region has been given by Smith et al. (2012). According to these authors the goal of 1% absolute accuracy in line intensities, as demanded by Flaud and Bacis (1998), is not yet attained.

Initiated by several more recent laboratory intercomparisons of ozone absorption coefficients in the mid-infrared and UV

spectral regions (Picquet-Varrault et al., 2005; Gratien et al., 2010; Guinet et al., 2010) a debate was reopened on whether the ozone line intensities in the 10 $\mu$m region, which in the MIPAS spectroscopy and in HITRAN version 2004 and later ones are lower by $\sim$4% than in HITRAN-1996, have to be increased by 3–5% again. This is also supported by an intercomparison of total ozone columns by Schneider et al. (2008), who found a bias of 4–5% between groundbased FTIR observations at 991–1007 cm$^{-1}$ and Brewer measurements. A review of laboratory and field studies related to this topic was given by Orphal

et al. (2016). However, rescaling of the ozone band intensities would only have an effect on the channel AB-A bias observed in MIPAS data, if the $\nu_2$ band used in channel A would not be scaled by the same amount as the bands in the 10 $\mu$m region applied in channel AB.

MIPAS measurements are performed in several channels covering the mid-infrared spectral region. Especially suited for MIPAS ozone retrieval are the strong fundamental $\nu_3$ band centered at 1042 cm$^{-1}$, but also the weaker fundamental $\nu_1$ and $\nu_2$

bands at 1103 and 701 cm$^{-1}$. While the $\nu_1$ and $\nu_3$ bands are mainly situated in MIPAS channel AB (1020–1170 cm$^{-1}$), the $\nu_2$ band is located in channel A (685–970 cm$^{-1}$). The fact that ozone retrievals using MIPAS channel AB (1020–1170 cm$^{-1}$) microwindows (MWs) are biased high by up to 8% compared to retrievals based on MIPAS channel A (685–970 cm$^{-1}$) MWs has already been reported by Glatthor et al. (2006) for measurements in the so-called high-resolution mode (2002–2004), who concluded that the major part of the differences results from spectroscopic inconsistencies. These retrievals were performed

using the ozone line list of the MIPAS database version pf3.2 provided by Flaud et al. (2003b). The bias between MIPAS channel A and AB retrievals is of particular importance, because ozone data retrieved at IMK from a combination of channel A and AB microwindows (versions V5R_O3_220 and V5R_O3_221) have been found to be biased high in the altitude region around 40 km (Laeng et al., 2014). Since the uncertainties in line intensity of many strong lines - especially in the $\nu_3$ band - have been declared to be less than 2% (Wagner et al., 2002), the problem has been reassessed, and various potential reasons

for the deviations have been examined.

In Sections 2-3 we shortly describe the MIPAS experiment and the retrieval setup, followed by a presentation of the ozone profiles resulting from retrievals using the MIPAS pf3.2 spectroscopy in Section 4. In Sections 5-6 we show investigations, which widely exclude forward modelling, instrumental or calibration issues as reason for the observed bias. In Section 7.1 we apply the ozone line data of various versions of the HITRAN database. In Section 7.2 we present a comparison between

retrievals using ozone lines of a completely different database, namely GEISA-2015 (Jacquinet-Husson et al., 2016), and

retrievals based on the HITRAN-2008 data. Then we demonstrate the possibility to reduce the channel AB-A differences by changing the air-broadening coefficients (Section 7.3). In Section 8 we show internal inconsistencies of air-broadening coefficients in the HITRAN-2008 database, followed by a summary and conclusions in Section 9.

## 2 Instrument description and retrieval setup

The Michelson Interferometer for Passive Atmospheric Sounding (MIPAS), which was operated on board the ENVIronmental SATellite (ENVISAT) between 2002 and 2012, has e.g. been described in Fischer et al. (2008). Therefore we give only a short description of the instrument. MIPAS was a limb-viewing Fourier transform infrared emission spectrometer covering the spectral region between 685 and 2410 cm$^{-1}$ (4.1–14.6 $\mu$m). From June 2002 to April 2004 MIPAS was operated in its original high resolution (HR) mode and since January 2005 in reduced resolution (RR) mode. We present retrievals based on data of the RR nominal measurement mode, which consisted of rearward limb-scans covering the altitude region between 7 and 72 km within 27 altitude steps. The level-1B radiance spectra used for retrieval are data version 5.02/5.06 and 7.11 (reprocessed data) provided by the European Space Agency (ESA) (Nett et al., 2002). The IMK notation for these spectra versions is V5R and V7R.

To reinvestigate the channel AB-A bias in retrieved ozone, retrievals using the processor of the Institut für Meteorologie und Klimaforschung and the Instituto de Astrofísica de Andalucía (IMK/IAA) were performed for 59 MIPAS orbits from January 3, April 1, July 2 and October 2-3, 2009. This processor uses the Karlsruhe Optimized and Precise Radiative Algorithm (KOPRA) (Stiller, 2000) for radiative transfer calculations and the Retrieval Control Program (RCP) of IMK/IAA for inverse modelling of spectra. Two microwindow setups were used, one in the spectral range of MIPAS channel A and the other in the range of channel AB (Tables 1, 2). Both of the setups consist of the large number of 30 MWs to obtain a high vertical resolution. The microwindows used in channel A span the wavenumber region 687–791 cm$^{-1}$ covering the fundamental $\nu_2$ band and those of channel AB the region 1028–1164 cm$^{-1}$ covering the fundamental $\nu_1$ and $\nu_3$ bands, respectively. The strong lines of the $\nu_1$ and $\nu_3$ bands are suited for ozone retrieval in the middle atmosphere, but especially the $\nu_3$ lines become saturated for limb scans through the ozone concentration maximum at $\sim$ 28 km and through the lower stratosphere, where the $\nu_2$ lines are a suitable alternative. Consequently, the first four microwindows covering the central part of the $\nu_3$ band are mostly omitted at these altitudes in the MW selection for channel AB retrievals because of saturation (Table 2). In addition to the lines of the fundamental bands, the spectral regions used for retrieval contain a large number of ozone lines from higher transitions.

Except of the use of dedicated microwindows restricted to MIPAS channels A or AB the setup is the same as for IMK retrieval version O3_V5R_220, consisting in a joint-fit of ozone, microwindow-dependent continuum profiles and a microwindow-dependent, but height constant spectral offset. Temperature-, pressure- and $H_2O$-profiles required for forward modelling were taken from preceding retrieval steps, and the profiles of the remaining interfering species were taken from the climatology provided by Remedios et al. (2007). For all but one retrieval tests presented here spectra of version V5R of the reduced spectral resolution period were taken. Since the subsequent set of MIPAS spectra (version V7R) presumably was produced using an improved calibration scheme, one additional channel AB-A intercomparison was performed on the basis of this data set. While

the spectroscopic data for ozone were changed in several tests, the same line lists were always used for all other gases. More information on trace gas retrieval from MIPAS data as performed at IMK can be found in various papers, e.g. in von Clarmann et al. (2003) or in Höpfner et al. (2004).

## 3 Error estimates of ozone line and band intensities

As specified by Flaud and Piccolo (J.-M. Flaud, pers.comm.) the relative error SX in line intensity of ozone lines of the fundamental $\nu_1$, $\nu_2$ and $\nu_3$ bands in the MIPAS database can be parameterized as follows:

$$SX = 0.02 \times (1 + JU/70 + KU/25), \tag{1}$$

where JU is the upper state rotational J quantum number and KU the upper state rotational K quantum number. For the other transitions ending at the ground state the error is

$$SX = 0.03 \times (1 + JU/60 + KU/20). \tag{2}$$

For higher transitions the relative error gradually increases up to

$$SX = 0.10 \times (1 + JU/35 + KU/11) \tag{3}$$

for the transitions ending at strongly excited lower states.

The error estimates for ozone line intensities in the HITRAN-2008 database are of similar magnitude (Rothman et al., 2009). In the region of the channel A microwindows they are 1–2% for the strongest lines, 2–5% for lines of medium strength and 5–10% for weak lines. For the lines in channel AB no error estimates are given for line intensities. The HITRAN-2008 error estimates for the air-broadened halfwidths vary between 2–5% for the strongest lines and 10–20% for weak lines.

Due to the high accuracy in line intensities as outlined in the Introduction, a potential systematic offset between the ozone bands in MIPAS channels A and AB can be assumed to be lower than 2%. This is also justified by the fact that one of the spectroscopic data sets that were used as basis for the ozone line lists in the MIPAS database as well as in all recent HITRAN compilations (version 2004 and later ones), consists in simultaneous laboratory measurements of the $\nu_1$, $\nu_2$ and $\nu_3$ band regions (Wagner et al., 2002).

## 4 Retrievals using the MIPAS spectroscopy

As already mentioned in the Introduction, a dedicated spectroscopic database for analysis of MIPAS data was established by Flaud et al. (2003b). Starting point for this database was the HITRAN-1996 edition. The new line lists were validated by

comparison between atmospheric simulations and ATMOS (Atmospheric Trace Molecule Spectroscopy) as well as MIPAS measurements. The focus of these investigations was a spectroscopic update of the main target species of MIPAS, namely $H_2O$, $CO_2$, $O_3$, $CH_4$, $NO_2$ and $HNO_3$. The ozone spectroscopy was updated on the basis of three sets of highly consistent experimental data (Flaud et al., 2003a). More specific, a new compilation of the fundamental $\nu_1$, $\nu_2$ and $\nu_3$ bands was created and complemented by all higher transitions contained in the HITRAN-1996 database. Since the line intensities in the fundamental $\nu_1$ and $\nu_3$ bands of the new data set were about 4% lower than the corresponding intensities in HITRAN-1996, the line intensities of all bands adopted from HITRAN-1996 were divided by 1.04.

Figure 1a shows average ozone profiles resulting from retrievals in MIPAS channels A and AB using the ozone linelist of the MIPAS database version 3.2, which has also been applied in an earlier investigation (Glatthor et al., 2006). This ozone line list is also applied for operational ozone retrieval at IMK/IAA. Averaging was performed over all geolocations of the 59 evaluated orbits (cf. Section 2). Similar as in the the previous investigation, use of the MIPAS spectroscopy leads to systematically higher ozone values using MIPAS channel AB microwindows as compared to channel A MWs. The absolute differences are largest in the height region 28–45 km and amount to 0.4 ppmv at 36 km altitude (Fig. 1b), which corresponds to a relative difference of 6% (Fig. 1c). This difference is larger than the relative error in line intensity given in Eqs. 1 and 2 for the strongest and medium scale ozone lines (at least for transitions with low to medium-sized rotational quanta JU and KU). The large scatter of the relative differences below 20 km is caused by the much lower ozone VMRs and by averaging of increasingly less values due to cloud filtering. For different latitude bands (Fig. 1d) or seasons (Fig. 1e) the relative differences in the altitude range 30 to 45 km vary between 5 and 7.5%.

The vertical resolution in terms of full width at half maximum (FWHM) in channel AB is up to 1 km worse than in channel A between 10 and 35 km altitude, nearly the same between 35 and 40 km and up to 0.5 km better above 40 km (Figs. 1f, 1g). This shows the somewhat better appropriateness of channel A microwindows for ozone retrieval in the lower stratosphere and of channel AB microwindows in the upper stratosphere and mesosphere. The effect of the differences in height resolution on the retrieved VMRs was tested by application of the averaging kernels from channel A retrievals to channel AB profiles and vice versa. This led to negligible changes of the profiles only (not shown).

The corresponding ozone profile of the most actual IMK ozone data version using V5R-spectra (V5R_O3_224.1) is practically identical to the channel A result presented here. The reason is a nearly identical set of microwindows used for the standard retrieval. The almost complete restriction to channel A microwindows (except of two channel AB MWs used above 50 km) resulted from a validation study (Laeng et al., 2014). This study had shown that the earlier ozone data version V5R_O3_220.1 was biased high in the altitude range 35–45 km due to a higher fraction of channel AB microwindows, applied at 36 km and above.

## 5 Exclusion of forward modelling issues

Similar ozone retrievals with microwindows situated in MIPAS channels A or AB were also performed with the Bologna Geo-fit Multitarget Retrieval Model (GMTR), which uses a different forward model (Carlotti et al., 2006). The differences

between channel A and AB retrievals are very similar to those resulting from the IMK/IAA retrievals (Figure 2). This agreement widely excludes the hypothesis that the bias is caused by deficiencies in the KOPRA forward model used at IMK. Nevertheless, a number of possible forward modelling issues has been investigated and is discussed below.

Channel-dependent accuracy parameters: To check if the higher ozone amounts retrieved in MIPAS channel AB are caused by disproportionately high rejection of weak lines by KOPRA in modelling of the absorption coefficients in this spectral region, channel AB retrievals were performed with strongly increased accuracy in calculation of the absorption coefficients. This leads to better consideration of weak lines. However, these retrievals resulted in nearly identical ozone profiles (not shown).

Channel-dependent continua: Since the continuum profiles fitted in the first four microwindows of the dedicated channel AB occupation matrix often exhibit unphysical negative excursions in the altitude region 35 to 40 km, additional channel AB retrievals were performed without these microwindows. The effect on the retrieved ozone profiles was negligible (not shown).

Non-local thermodynamic equilibrium (NLTE) effects: Another possible reason for the bias in ozone VMRs could be different
strengths of NLTE effects in MIPAS channels A and AB. Like for standard MIPAS ozone retrievals from nominal mode measurements, time consuming modelling of NLTE effects was not taken into account for the channel A and AB retrievals. This is generally justified, because these effects are mostly small in the stratosphere, and spectral regions subject to NLTE effects are avoided in the microwindow selection. Nevertheless, channel A and AB ozone retrievals including modelling of NLTE effects had been performed by Glatthor et al. (2006), which had shown that neglect of NLTE modelling is not
the dominant reason for the channel AB-A bias. Such calculations were not repeated here. Instead, since NLTE conditions generally persist during daytime, averaging was performed separately for day- and nighttime profiles of the data set (Figure 3). It is evident that the channel AB-A differences are nearly the same for the whole data set as well as for the day- and nighttime measurements. According to this estimation, NLTE effects have only little influence on the observed differences.

Non-Voigt line shape: As commonly done in radiative transfer calculations for spaceborne mid-IR measurements, line modelling with KOPRA is performed assuming a Voigt line shape. This assumption is confirmed by Tran et al. (2010), who showed that for the entire $10\mu$m ozone band non-Voigt line shape effects, represented by a speed dependent Voigt model, lead to errors in retrieved atmospheric ozone of less than 1%. These investigations were based on calculated as well as on measured spectra obtained by limb-viewing solar occultation and emission measurements. Therefore we conclude that the channel AB-A bias
for the most part can not be explained by neglect of non-Voigt effects.

## 6  Exclusion of instrumental and calibration issues

To exclude instrumental or calibration issues, ozone retrievals using channel A and AB microwindows were also performed for measurements of the MIPAS-balloon instrument (Friedl-Vallon et al., 2004), i.e. for a completely independent experiment

with different level-1 processing and calibration procedures (Figure 4). The balloon spectra were obtained on 31 March 2011 over Esrange, Sweden (67.9°N, 21.1°E) and on 14 June 2005 over Teresina, Brazil (5.1°S, 42.9°W). These retrievals resulted in channel AB–A differences of 0.5 ppmv in the altitude range of 30–40 km, which are similar to those found in the spaceborne MIPAS observations. This agreement largely excludes inconsistencies in calibration of channel A and AB spectra, different detector alignment or instrumental line shape issues in the spaceborne MIPAS data. Nevertheless, instrumental and calibration issues have been investigated, and the most important ones are shortly discussed below.

Spectral calibration issues: To check deficiencies in spectral calibration, ozone retrievals were also performed with MIPAS spectra version V7R, which had been generated with an improved calibration scheme. However, this test resulted in even somewhat larger channel AB-A differences than retrievals with MIPAS V5R spectra (cf. Figure 5).

Line of sight issues: The vertical field-of-view of the MIPAS experiment is assumed to be the same for each channel. However, due to detector misalignment the effective line of sight might vary between the different MIPAS channels. Although a detector misalignment is widely excluded by the similarity of the MIPAS-balloon results, this problem has been investigated. It turned out that the field-of-view of channel AB would have to be shifted upward by more than 500 m to remove the channel AB-A differences in the height region 33-40 km (not shown). This shift is much larger than the instrumental requirement for inter-channel co-alignment, which is 1.3 mdeg or 68 m, and therefore rather unrealistic. Thus, this investigation confirms that inter-channel misalignment is not the cause of the channel AB-A bias.

## 7    Retrievals using additional spectroscopic databases

Since deficiencies in forward modelling, instrumental and calibration issues can largely be ruled out as reason for the bias between channels A and AB, inconsistencies in spectroscopic data practically are the only explanation left. For this reason we performed additional ozone retrievals using different HITRAN versions and the GEISA-2015 database to check if there is any line list, which produces more consistent ozone profiles for channel A and AB retrievals.

### 7.1    Comparison of different HITRAN-versions

Figure 5 shows average ozone profiles and channel AB-A differences for retrievals using ozone linelists of versions 1996, 2004, 2008 and 2016 of the HITRAN spectroscopic database (Gordon et al., 2017, and references therein) as well as of the MIPAS spectroscopy. Retrievals performed with ozone line data of HITRAN versions 2000 and 2012 are not shown, because these linelists are practically identical to HITRAN-1996 and to HITRAN-2008, respectively. HITRAN versions 2004, 2008 and 2016 lead to nearly the same channel A (Fig. 5a) and channel AB profiles (Fig. 5b) as the MIPAS spectroscopy. Therefore these HITRAN versions also result in almost the same channel AB-A differences (Figs. 5c,d). For each of these databases the maximum difference is 0.5 ppmv (7%) at the altitude of 36 km, which is even somewhat larger than the bias resulting from the MIPAS spectroscopy. Consequently, these differences are also larger than the relative errors in line intensity given in Eqs. 1

and 2. Only the HITRAN-1996 spectroscopy leads to a somewhat different result, namely positive channel AB-A differences of up to 0.5 ppmv above 32 km, but negative differences of up to -0.4 ppmv below this altitude. Similar results in case of use of the HITRAN-1996 spectroscopy have already been shown by Glatthor et al. (2006). Since the correction scheme of detector nonlinearities is assumed to be improved in generation of MIPAS V7R spectra, an additional retrieval test was performed using this dataset and HITRAN 2008 line data. However, these spectra lead to even somewhat larger differences (yellow curve) than the V5R spectra.

The rather good agreement between the different channel A as well as channel AB retrievals indicates largely consistent spectroscopic parameters of corresponding ozone lines in the MIPAS spectroscopy and the HITRAN-2004 database as well as in later HITRAN versions for the spectral range of the channel A and AB microwindows. Therefore a comparison between the line parameters of the MIPAS and HITRAN databases is not presented here, but provided as supplemental material only.

## 7.2 HITRAN-2008 versus GEISA-2015

Since the channel AB-A bias could not be removed by use of any of the HITRAN line lists, we performed an additional retrieval test with a different spectroscopic database, namely the Gestion et Etude des Informations Spectroscopiques Atmosphériques - version 2015 (GEISA-2015) compilation (Jacquinet-Husson et al., 2016) and compared the results with those based on HITRAN-2008. Although the line parameters of the three fundamental ozone bands in GEISA-2015 are principally obtained from the same sources as those in HITRAN-2008 (Jacquinet-Husson et al., 2008, and references therein), we found considerable differences.

### 7.2.1 Retrieval results

Figure 6 shows average ozone profiles resulting from retrieval in MIPAS channels A and AB using the ozone line lists of HITRAN-2008 (cf. Figure 5) and of GEISA-2015. The GEISA-2015 database leads to nearly the same profile as HITRAN-2008 for retrievals using the channel AB microwindows, but to even higher ozone VMRs for retrievals using the MWs in channel A, i.e. to a reversal of the bias obtained with the HITRAN line data. The differences between channel AB and A profiles obtained with the GEISA-2015 spectroscopy are negative in the height region 20–43 km, amounting up to -0.55 ppmv or -7.5% at 28 km altitude. Moreover, the differences between the channel A retrievals, which definitely have spectroscopic reasons, are even larger and amount up to 0.8 ppmv or about 10%. With regard to the validation study by Laeng et al. (2014) the channel A profiles obtained with the GEISA-2015 spectroscopy are most probably biased high.

### 7.2.2 Comparison of spectral parameters

The retrieval results indicate mostly consistent spectral parameters in HITRAN-2008 and GEISA-2015 for the ozone lines used in MIPAS channel AB, but considerable spectroscopic differences in the region of the channel A microwindows. In the following, we will compare the HITRAN-2008 and GEISA-2015 ozone lines applied in channel A as well as in channel AB to identify the parameters responsible for these differences.

First of all we compared the number of ozone lines of the two databases in the spectral region covered by the channel A microwindows. In this wavenumber region the GEISA-2015 database contains 3631 lines, all of them having a corresponding line in the HITRAN-2008 edition. The latter contains 734 additional lines, which however are rather weak. Their intensities are between $2.013\times10^{-26}$ and $5.858\times10^{-24}$ cm$^{-1}$/(molecules cm$^{-2}$), while a considerable number of the lines contained in both databases have intensities between $1\times10^{-22}$ and $1.66\times10^{-21}$ cm$^{-1}$/(molecules cm$^{-2}$). A test retrieval without these additional lines (not shown) resulted in nearly the same ozone profiles as the retrieval using the complete HITRAN-2008 ozone linelist. In the spectral range of the channel AB microwindows the GEISA-2015 and HITRAN-2008 databases contain 3737 and 3804 lines, respectively. The number of corresponding lines is 3724, i.e. the HITRAN-2008 edition contains 80 lines, which are not in the GEISA-2015 compilation. Again, these lines have intensities below $9.21\times10^{-25}$ cm$^{-1}$/(molecules cm$^{-2}$) only, while a lot of the lines, which are available in both databases, have much higher intensities between $1\times10^{-22}$ and $3.9\times10^{-20}$ cm$^{-1}$/(molecules cm$^{-2}$). In summary, the strong lines which make sizable contributions to the spectra are included in both data sets, and the missing weak lines are ruled out as cause of the discrepancy under investigation.

The spectral parameters which have the largest potential to cause the disagreement in channel A are line positions, line intensities or air-broadened halfwidths. Further parameters required for line modelling are lower state energies, the coefficients of temperature dependence of the air-broadened halfwidths and air-broadened pressure shifts. A comparison shows that the line positions and lower state energies of all GEISA-2015 and HITRAN-2008 lines inside the channel A microwindows agree exactly. Inside the channel AB MWs these parameters differ slightly between the two databases, but for a small number of very weak lines only. Compared to air-broadening, the relative contribution of self-broadening is of the order of $10^{-5}$ only and thus negligible. The pressure shift of the ozone lines in GEISA-2015 and most of the HITRAN-2008 lines used in channels A and AB is zero. About one third of the HITRAN-2008 lines have weak shifts of only -0.0008 and of -0.0007 cm$^{-1}$ in channel A and AB, respectively. Thus, differences in pressure shift are also negligible. The remaining parameters are line intensities and air-broadened halfwidths.

Figure 7 shows the relative differences between the intensities of the ozone lines of the GEISA-2015 and the HITRAN-2008 database used for channel A and AB retrievals. It is evident that the intensities of the strongest lines used in channel A are practically identical. There are slight deviations of 5% for a part of the weaker lines only. This indicates that the differences in channel A retrievals do not result from inconsistent line strengths. The respective difference plot of the lines used in MIPAS channel AB also shows nearly identical intensities of the strongest lines. Again there are differences of +5% for weaker lines and larger differences of up to $\pm25\%$ for very weak lines. However, the good agreement of the channel AB profiles (Figure 6) shows, that a potential influence of the different line strengths of the weak lines is low.

Next we correlated the air-broadened halfwidths $\gamma_{air,0}$ of the ozone lines used for channel A and AB retrievals of the two databases, colour-coded by the respective line strengths (Figure 8). In both cases the air-broadened halfwidths of the HITRAN-2008 ozone lines are larger than those of the GEISA-2015 compilation, but there are clear systematic differences between the channel A and channel AB correlations for the strongest lines. In channel AB the air-broadened halfwidths of the strongest lines are largely identical. However in channel A the $\gamma_{air,0}$-values of the strongest lines are significantly lower ($\sim13\%$) in the

GEISA-2015 database. Differences between line broadening parameters in the GEISA and HITRAN databases have already been reported by Jacquinet-Husson et al. (2008, 2011).

The air-broadened halfwidths $\gamma_{air,0}$ given in the line databases are reference values for $p_0 = 1013.25$ hPa and $T_0 = 296$ K. The $\gamma_{air}$-values for actual temperature $T$ and pressure $p$ are calculated as follows:

$$\gamma_{air}(p, T) = \gamma_{air,0} \times p/p_0 (T_0/T)^n. \tag{4}$$

The coefficients $n$ of temperature dependence are also given in the spectroscopic databases. Figure 9 shows a scatter plot of the coefficients $n$ of GEISA-2015 versus HITRAN-2008, again colour-coded by line strength. In channel A the coefficients $n$ of the strongest lines are nearly equal for both databases. Thus there is no compensation of the large differences in the air-broadened halfwidths of channel A via the coefficients $n$. In channel AB there are somewhat larger differences of some of the coefficients $n$ belonging to the strongest lines, namely the coefficients at the ordinate value of 0.76. However, since the channel AB profiles are nearly identical, these deviations obviously have no large effect. For example, shifting of the HITRAN-2008 values of 0.71 of several strong lines (red pluses) to the main diagonal changes $\gamma_{air}$ by 1.3% only for a stratospheric temperature of 230 K.

The correlation analysis gives strong indication that the differences of the ozone VMRs resulting from retrieval in channel A using HITRAN-2008 or GEISA-2015 data are not caused by differences in line strengths but rather by the differences between the air-broadened halfwidths of the strongest lines. To check this assumption we replaced the $\gamma_{air,0}$-values and the coefficients $n$ of temperature dependence in the GEISA-2015 database by the respective HITRAN-2008 values and performed additional retrievals using these line parameters. Figure 10 shows the channel A and AB retrieval results using the original and the modified linelist. After modification of the GEISA-2015 data the average ozone profile retrieved in channel A is nearly identical to the profile resulting from the HITRAN-2008 spectroscopy. This result confirms that the bias in channel A retrievals between use of the HITRAN-2008 and of the GEISA-2015 database is caused by the differences in air-broadened halfwidths. Furthermore, the small differences between the retrievals in channel AB are even more reduced.

### 7.3 Channel AB retrievals using modified HITRAN-2008 lines

Since the relative differences between the intensities of the ozone bands in MIPAS channels A and AB are assumed to be very small (1-2%), we posit that a considerable part of the database-internal channel AB-A biases might be caused by inconsistent $\gamma_{air,0}$-values as well. We performed two additional tests to check if the differences can be reduced by modification of the air-broadened halfwidths. In the first test we increased the halfwidths of the HITRAN-2008 ozone lines in the spectral region of MIPAS channel AB by 0.005 cm$^{-1}$/atm@296K (5–7%). Caused by this change the deviations in the height range 32–45 km are considerably reduced as compared to the retrieval using the unmodified HITRAN-2008 database, namely to 0.15 ppmv or less (Figure 11). On the other hand there are now larger negative deviations of up to -0.2 ppmv in the height range 18–32 km. In the second test we increased the halfwidths of the strongest lines on the main diagonal in Figure 8 (right) by 0.008 cm$^{-1}$/atm@296K to obtain a similar distribution like the channel A correlation in Figure 8 (left). This led to even lower differences between 18 and 32 km, but to somewhat larger negative differences between 18 and 32 km.

As a side aspect, Figure 11d shows the differences between the channel AB retrievals with increased air-broadened halfwidths and the channel AB retrievals using the unmodified HITRAN-2008 spectroscopy. Since these differences are relatively constant around -4 and -5% over the altitude region 10–40 km, they are good estimates of the respective changes in ozone column amounts. Thus, this result shows that, alternatively to the proposed re-scaling of line intensities in the $10\mu m$ region (cf. Section 1), the bias between ozone column amounts measured in the mid-infrared and UV spectral regions as e.g. shown by Schneider et al. (2008) could probably also be reduced by change of the air-broadened halfwidths.

## 8   Additional investigations

To substantiate the possibility of inconsistent $\gamma_{air,0}$-values, Figure 12 shows modelled ozone spectra using the HITRAN-2008 and the MIPAS spectroscopy for a tangent altitude of 30 km. In the difference plot (Fig. 12c) a number of conspicuous lines belonging to the fundamental $\nu_2$ band can be identified between 797 and 830 cm$^{-1}$ (note that these lines are not contained in our set of channel A microwindows). The line intensities of these transitions are identical in both spectroscopic line lists and thus not responsible for the deviations between the model spectra. However, there are substantial differences between the air-broadened halfwidths (Figure 13). The halfwidths of the lines of the MIPAS database continuously decrease between 700 and 850 cm$^{-1}$. The HITRAN-2008 values are slightly larger, but decrease in a comparable manner up to $\sim$790 cm$^{-1}$. However, there is a strong increase of more than 0.01 cm$^{-1}$/atm@296K at $\sim$797 cm$^{-1}$, leading to significantly larger values up to 850 cm$^{-1}$. This jump in $\gamma_{air,0}$ is the reason for the stronger ozone lines in the model spectrum using HITRAN-2008 data in Figure 12. This artefact is still present in later versions up to HITRAN-2016. In addition to the spikes at and above 797.05 cm$^{-1}$ the difference plot (Fig. 12c) also exhibits somewhat smaller peaks for the $\nu_2$ transitions at lower wavenumbers, e.g. at 789.11, 781.18, 773.29 and 765.43 cm$^{-1}$. This indicates potential additional spectroscopic inconsistencies below 797 cm$^{-1}$ in the HITRAN-2008 database.

To further demonstrate the inconsistencies of the spectroscopic parameters of the lines above 790 cm$^{-1}$ in the HITRAN-2008 data base identified above, broad-band ozone retrievals were performed in the wavenumber range 795 to 825 cm$^{-1}$ for MIPAS orbits 39680–39693. In addition to ozone, peroxyacetyl nitrate (PAN), CCl$_4$, ClONO$_2$ and HCFC-22 were jointly fitted. The spectral signatures of additional gases were modelled by use of pre-fitted or climatological profiles. Figure 14 shows the residuals between mean modelled and measured spectra using the HITRAN-2008 and the MIPAS pf3.2 ozone spectroscopy for the latitude region 30°S–30°N and tangent heights covering the altitude region of $\sim$14–44 km. For the major part of the spectral region the residuals resulting from both spectroscopies are very similar, leading to a nearly complete coverage of the HITRAN-2008-residual. The large "blue" residuals are mainly caused by inadequately modelled CO$_2$ and H$_2$O lines. However, for tangent heights of 14, 20 and 30 km use of the HITRAN-2008 spectroscopy leads to large residuals (red) at the positions of the lines with the higher $\gamma_{air,0}$-values (797.05, 805.02 and 812.99 cm$^{-1}$). The spectroscopic parameters of these lines are not consistent to those of all the other ozone lines in the broad spectral range used for retrieval, and thus these lines can not be fitted properly. At 44 km these residuals have disappeared, showing that the influence of different air-broadened halfwidths becomes negligible at this altitude.

At first glance it seems a bit strange that increased air-broadened halfwidths lead to modelled ozone lines which exhibit stronger peak radiances (Figure 12). To investigate this behaviour more in detail, model calculations with KOPRA were performed for an isolated ozone line. For a homogeneous, optical thin atmosphere larger $\gamma_{air,0}$-values lead to lower radiances in the line center and to higher radiances at the line wings both for the monochromatic line and after folding with the apodized internal line shape (AILS) (not shown). However, for a limb observation through the Earth's atmosphere the result becomes different. For a tangent altitude of 30 km a larger air-broadened halfwidth also leads to a larger halfwidth and lower peak radiance of the modelled monochromatic line (Fig. 15a), but there is a significantly larger growth in radiance at the wings as compared to the decline in the line center (Fig. 15c). In this case, convolution with the AILS as performed in MIPAS retrievals indeed leads to a higher peak radiance (Figs. 15b,d).

## 9    Summary and conclusions

We have reassessed the bias in the altitude range 30–45 km between ozone retrievals using microwindows in MIPAS channels A (685–970 cm$^{-1}$) and AB (1020–1170 cm$^{-1}$). We found that the bias, originally detected in retrievals using V3O-spectra of the MIPAS high-resolution measurement period and the so-called MIPAS spectroscopy, also occurs for retrievals using later versions of level-1B spectra (V5R, V7R) of the MIPAS reduced-resolution mode. The effect amounts up to 8% at the altitude of 36 km. Forward modelling issues as reason for the problem could be excluded by the fact that similar differences also resulted from retrievals using the processor of the University of Bologna. Spectral calibration or line-of-sight issues could largely be excluded, because retrieval results of a different experiment (MIPAS-balloon spectra) also resulted in similar differences. Nevertheless, a number of forward-modelling, instrumental and calibration issues were examined, but did not lead to an explanation. The most plausible explanation left was inconsistencies in spectroscopic data.

Therefore additional retrievals were performed using several editions of the HITRAN database, which however led to rather similar, even somewhat larger channel AB-A differences in retrieved ozone as those resulting from application of the MIPAS line list. One exception is the HITRAN-1996 edition, which causes positive differences above, but negative differences below 32 km. Since the channel AB-A bias did not disappear by application of any of the HITRAN databases, we performed another retrieval test with a different spectroscopic database, namely the GEISA-2015 compilation. For measurements in MIPAS channel AB, the resulting ozone profiles are rather similar to those based on the HITRAN-2008 spectroscopy. However, for channel A retrievals the stratospheric ozone VMRs based on GEISA-2015 are up to 0.8 ppmv or about 10% larger than those based on HITRAN-2008 and even larger than the VMRs retrieved in channel AB. According to the results of the validation study by Laeng et al. (2014) these VMRs are obviously too high. We showed that the differences in channel A retrievals are not caused by inconsistent line intensities, but by ~13% lower air-broadened halfwidths of the strongest ozone lines in the channel A microwindows in the GEISA-2015 database. Since the relative differences between band intensities in the spectral ranges used in MIPAS channels A and AB are assumed to be considerably lower than the observed bias of 6–8%, we suggest that a major part of the channel AB-A differences might be caused by inconsistencies in air-broadened halfwidths in the individual line databases as well. To substantiate this assumption we identified several ozone lines in the HITRAN-2008 database, which

exhibit too large air-broadened halfwidths. The bias between channel A and AB can partly be reduced by increasing the air-broadened halfwidths of the lines in channel AB. In summary, the air-broadened halfwidths of ozone lines in the spectral regions of MIPAS channel A as well as of channel AB (i.e. the regions of the $\nu_2$, $\nu_1$ and $\nu_3$ fundamental bands) should be reassessed both for the GEISA and for the HITRAN databases. This is especially necessary for the GEISA-$\nu_2$ lines in MIPAS

channel A.

According to the investigations with our microwindow datasets, for the time being the best choice of an ozone line data compilation for evaluation of MIPAS measurements is the MIPAS spectroscopy (Flaud et al., 2003b), because on the one hand the channel AB-A differences are somewhat smaller than those resulting from the HITRAN databases and on the other hand it does not contain the inconsistency in some air-broadened halfwidths identified in HITRAN-2008, which was also transferred to

later HITRAN versions. However, as far as ozone is concerned we recommend to use version pf3.2 of the MIPAS spectroscopy and not the latest update pf4.45 (Flaud, et al., 2015, http://atmos.difa.unibo.it/spectdb/), because the ozone data set in this compilation is identical with HITRAN-2008.

*Acknowledgements.* The authors like to thank the European Space Agency for providing access to MIPAS level-1 data. This study was partly supported by the ESA CCI-O3 project. Meteorological analysis data were provided by the European Centre for Medium-Range Weather

Forecasts. We acknowledge support by the Deutsche Forschungsgemeinschaft and the Open Access Publishing Fund of the Karlsruhe Institute of Technology. Further, we thank F. Hase for helpful discussions.

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

**Table 1.** Microwindow set used for ozone retrieval in MIPAS channel A. The left column contains the spectral ranges of the microwindows. The matrix in the right column shows, at which height a particular microwindow is used (T) or not used (=). Heights increase from left to right, and height labels 06 to 75 km are to be read vertically in the first two rows.

| Microwindow cm$^{-1}$ | Altitude coverage 6–75 km |
|---|---|
| | 0 0 1 1 1222333344455566 6 6 7 7<br>6 9 2 5 8147036925814703 6 9 2 5 |
| 687.6875–688.6875 | =========T=TTTTTTTTTTTTT |
| 689.3125–691.8750 | ======TTT==TTTTTTTTTTTTT |
| 692.2500–695.1875 | ==========TTTTTTTTTTTTTT |
| 707.1250–710.0625 | =======TTTTTTTTTTTTTTTTT |
| 712.3125–713.4375 | TTT=======TTTT=T======= |
| 713.5000–716.4375 | TTT===TTTTTTTTT=TTTTTTT |
| 716.5000–719.4375 | =T====TT=TTTTTTTTTTTTTT |
| 720.7500–723.6875 | ==T====TTTTTTTTTTTTTTTT |
| 728.5000–729.3750 | TT======TTTTTTTTTTTTTTT |
| 730.0625–730.5000 | =TT=====TTTTTTTTTTTTTTT |
| 731.9375–732.8750 | TT====TTT==T===TTTTT=TT |
| 734.0000–734.7500 | TT======TT=TTTTT====TTT |
| 736.4375–739.3750 | TT==TT=T=TTTTTTTTTTTTTT |
| 739.4375–741.9375 | TT======TTTTTTTTTTTTTTT |
| 745.2500–745.6875 | T=TTT====TTTTTTTTTTTTTT |
| 746.6875–747.1250 | TT=T=TTTTTTTTTTTTTTTTTT |
| 747.6250–748.3750 | T======TTTT=TTTTT===TTTT |
| 749.5625–752.5000 | T==T=TTTTTTTTTTTTTTTTTT |
| 752.9375–755.8750 | ====TTT==TTTTTTTTTTTTTT |
| 758.3750–759.4375 | ===TTTTTTTTTTTTTTTTTTTT |
| 759.5000–761.8750 | TTTT==T==TTTTTTTTTTTTTT |
| 765.0000–765.6250 | TTTT======TTTTTTTTTTTTT |
| 767.5000–768.0000 | =TTTTTTTTTTTTTTTTTTTTTT |
| 771.8750–772.1250 | T=TTTTTTTTTTT==TTTTTTTT |
| 774.2500–774.5625 | T==T=TTT==TTTT==T===TT== |
| 776.5000–776.7500 | T===========TT=T=TTTTT== |
| 780.2500–781.9375 | TTTTT====TTTTTTTTTTTTTT |
| 788.9375–789.6875 | TTTTT====TTTTTTTTTTTTTT |
| 790.7500–791.0000 | T=TTT========T========= |
| 791.1875–791.5625 | TTT=TT====TTTTTTTTTTTTT |

**Table 2.** Microwindow set used for ozone retrieval in MIPAS channel AB. The left column contains the spectral ranges of the microwindows. The matrix in the right column shows, at which height a particular microwindow is used (T) or not used (=). Heights increase from left to right, and height labels 06 to 75 km are to be read vertically in the first two rows.

| Microwindow cm$^{-1}$ | Altitude coverage 6–75 km |
|---|---|
| | 0 0 1 1 1222333344455566 6 6 7 7 |
| | 6 9 2 5 8147036925814703 6 9 2 5 |
| 1028.6875–1031.3750 | =TT======TT====TTTTTTTTT |
| 1037.9375–1040.8750 | TT=======TT===TTTTTTTTTT |
| 1044.6250–1044.9375 | ============TTTTTTTT===T |
| 1050.4375–1050.7500 | ==========TT=TTTTTTTTTTT |
| 1070.4375–1070.7500 | T==TTTTTTT====TT==TTTTTT |
| 1072.0000–1072.3125 | T=T==TTTTTTTTTTT==TT=TT |
| 1073.6250–1074.4375 | T=====TTTTTTT=====TTTTTT |
| 1074.7500–1075.8750 | =========TTTTTTTTTTTTTTTT |
| 1077.2500–1078.4375 | ====TTTTTTTTTTTTTTTT==T |
| 1078.5000–1079.1250 | =T=TTT=======TTTTT==T=== |
| 1079.1875–1079.5625 | =TTTTTTTTTTTTTTTTTTTTTTT |
| 1081.1250–1083.4375 | T====T==TTTTTTTTTTTTT=== |
| 1115.8125–1116.2500 | =====TTTTTTTTTTTTTTTTTTT |
| 1117.1875–1117.4375 | TTTT==TTTTT=T==TTTTTTTTT |
| 1118.1250–1118.4375 | TT=TTT===TTTTTT=TTTTTTTT |
| 1119.6250–1119.8750 | TTTTT=TTT=TTT==TTTT====T |
| 1122.6875–1123.1250 | TTTTTTTTTTTTTTTTTTTTTTTT |
| 1124.8750–1125.1875 | T====TTTTTTTTTTTTTTTTTTT |
| 1126.1875–1126.4375 | =TTTTTTTT=============== |
| 1127.2500–1127.5000 | TTT=T================= |
| 1127.8750–1128.3125 | TTTTT===TTTT==TTTTTTTTTT |
| 1128.4375–1128.8125 | TTTTTTTT=TTT===TTTTT==TT |
| 1129.3125–1129.7500 | TTTTTTTTTTTT===TTTTTTTTT |
| 1131.2500–1134.1875 | T=TT==TTTTTTTTTTTTTTTTTT |
| 1139.6875–1140.0625 | TT=T=TTTTTTTTTTTTTTTTTTT |
| 1140.8750–1141.1250 | TTTTTTT==TTTTT=TTTTT===T |
| 1160.5625–1160.8125 | =TT=====TT=TTT=======TT= |
| 1160.9375–1161.3750 | TTTTTTTTTTT===TTTTTTT==T |
| 1161.4375–1161.7500 | TTTTTTTTTT============== |
| 1163.7500–1164.0625 | =TT===TTTTTTTTTTTTTTTTTT |

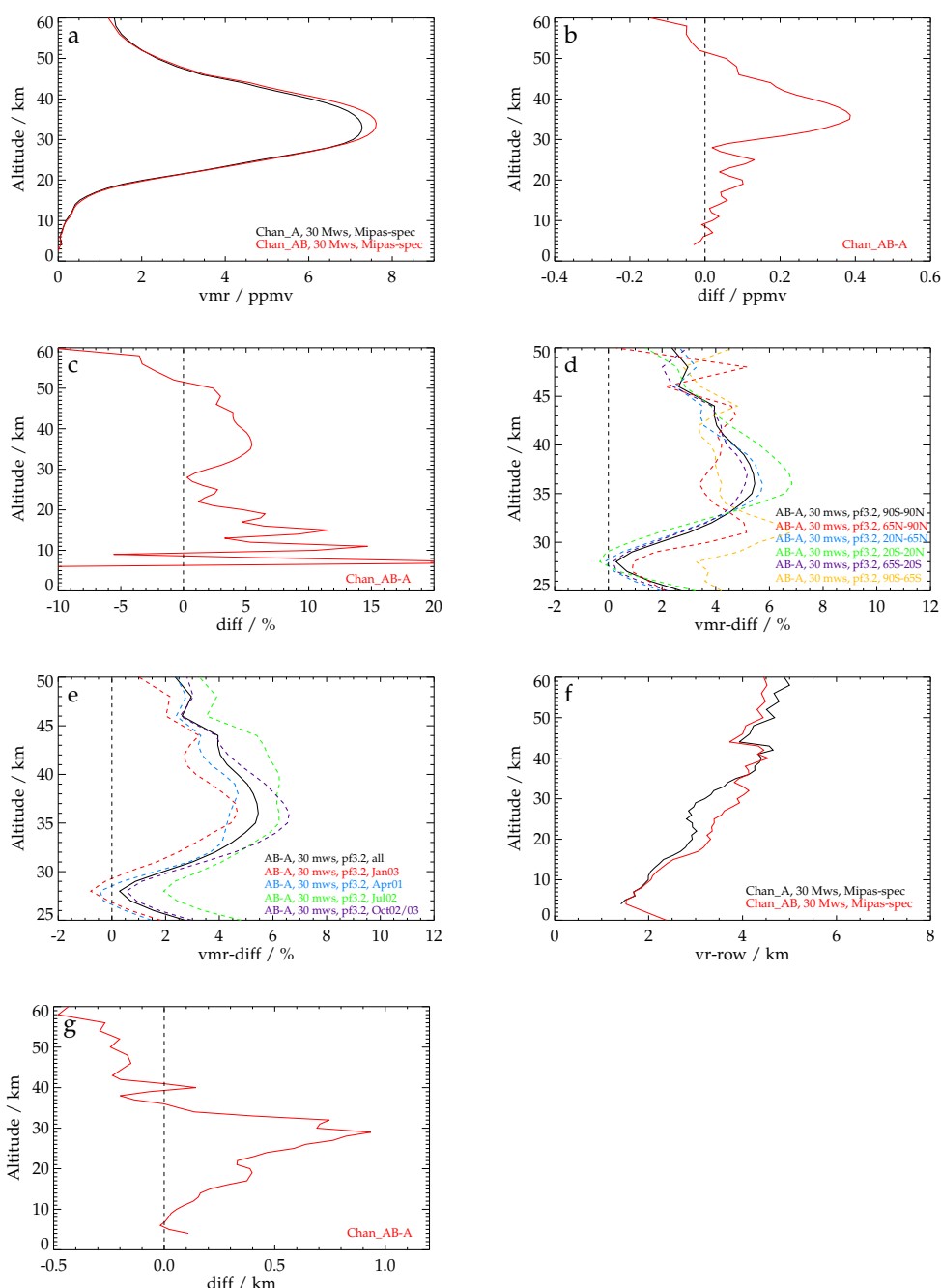

**Figure 1.** (a) Mean ozone profiles retrieved in MIPAS channels A (black) and AB (red) using the MIPAS pf3.2 spectroscopy. Averaging was performed over all profiles of 59 orbits from January 3, April 1, July 2 and October 2/3, 2009. (b) Absolute and (c) relative differences between the profiles retrieved in channels AB and A. (d) Relative AB–A differences for all data (solid line) and in the latitude bands 65–90N, 20–65N, 20S–20N, 65–20S and 90–65S (dashed lines, see legend). (e) Relative AB–A differences for all data (solid line), January 3, April12, July 2 and October 2/3, 2009 (dashed lines, see legend). Note the different height scale for the latitudinal and seasonal display. (f) Mean vertical resolution of the MIPAS channel A (black) and AB retrievals (red). (g) Respective difference.

## Global Average Orbits 02633+02634          Averaged Differences

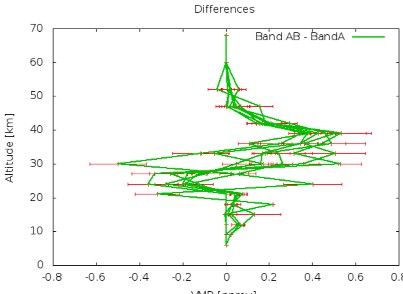

**Figure 2.** Left: Mean ozone profiles of MIPAS orbits 2633 and 2634 retrieved with the Bologna Geo-fit Multitarget Retrieval Model (GMTR) using microwindows in MIPAS channels A (red) and AB (blue). Right: Respective AB–A differences for several latitude bands.

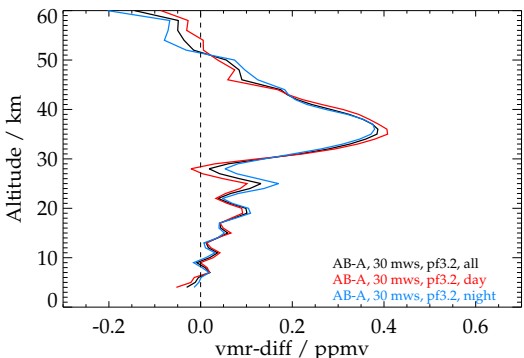

**Figure 3.** Difference between mean ozone profiles retrieved in MIPAS channels AB and A using the MIPAS pf3.2 spectroscopy. Averaging was performed over 59 orbits from January 3, April 1, July 2 and October 2/3, 2009, for all (black), daytime (red) and nighttime profiles (blue).

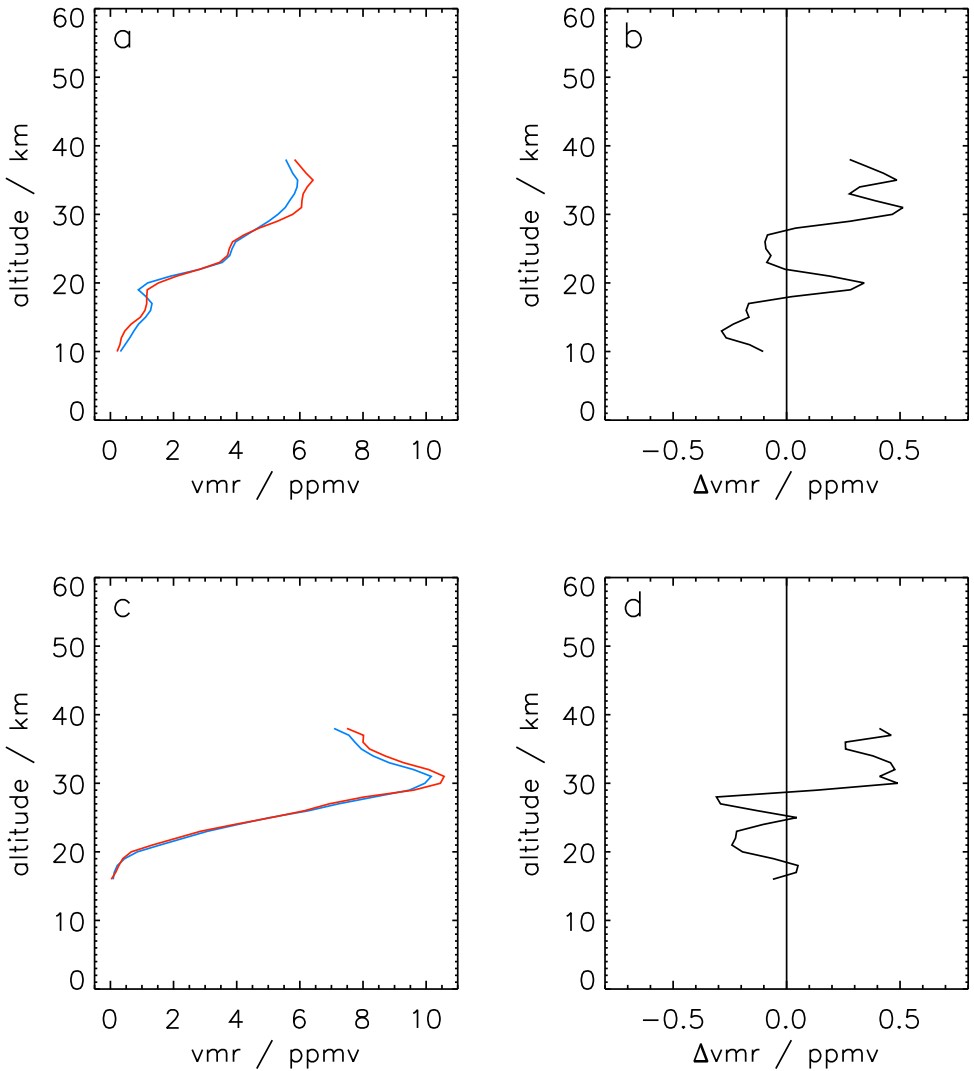

**Figure 4.** (a) Ozone profiles retrieved from MIPAS-balloon measurements above Esrange (67.9°N, 21.1°E), Sweden, on 31 March 2011, using microwindows in MIPAS channels A (blue) and AB (red). (b) Respective difference. (c) Same as (a) but for MIPAS-balloon measurements above Teresina (5.1°S, 42.9°W), Brazil, on 14 June 2005. (d) Respective difference.

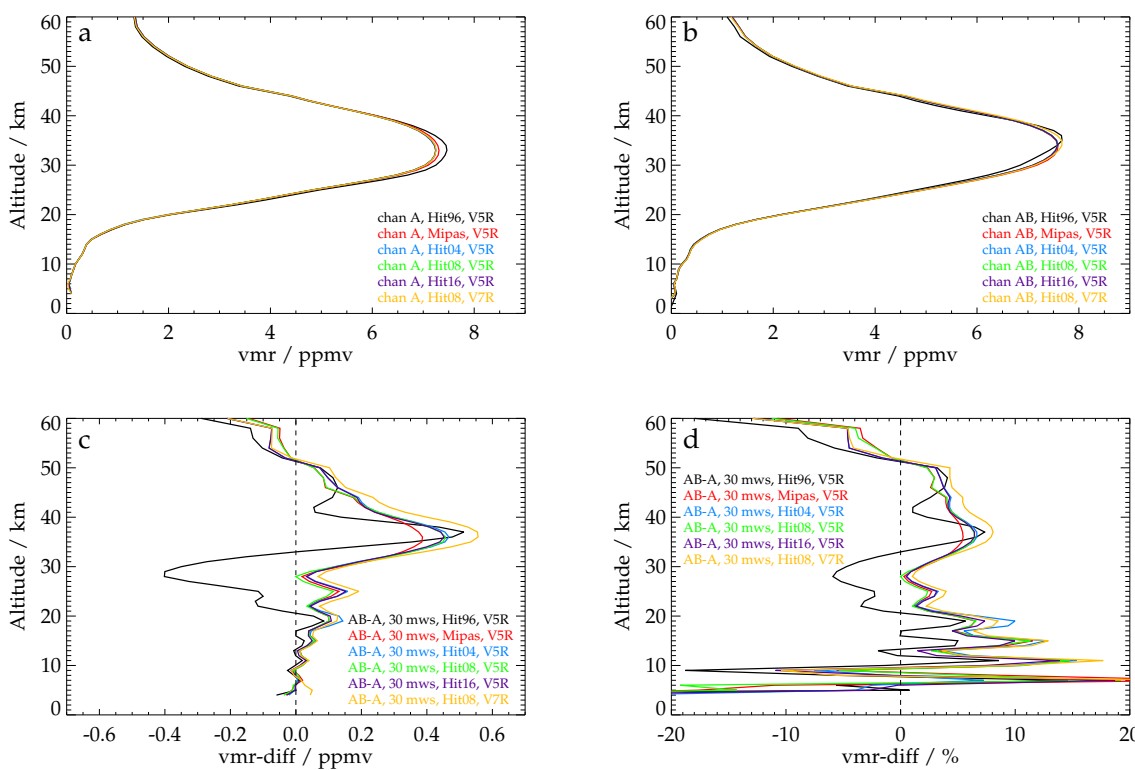

**Figure 5.** (a) Average ozone profiles retrieved in MIPAS channel A using V5R spectra and different spectroscopies: HITRAN-1996 (black), MIPAS pf3.2 (red), HITRAN-2004 (blue), HITRAN-2008 (green) and HITRAN-2016 (violet). The yellow curve shows the average profile for V7R spectra and application of the HITRAN-2008 line list. (b) Same as A, but for MIPAS channel AB. (c) Absolute and (d) relative differences between the profiles retrieved in channels AB and A. The ozone profiles were averaged over all single-scan profiles of 59 orbits from January 3, April 1, July 2 and October 2/3, 2009.

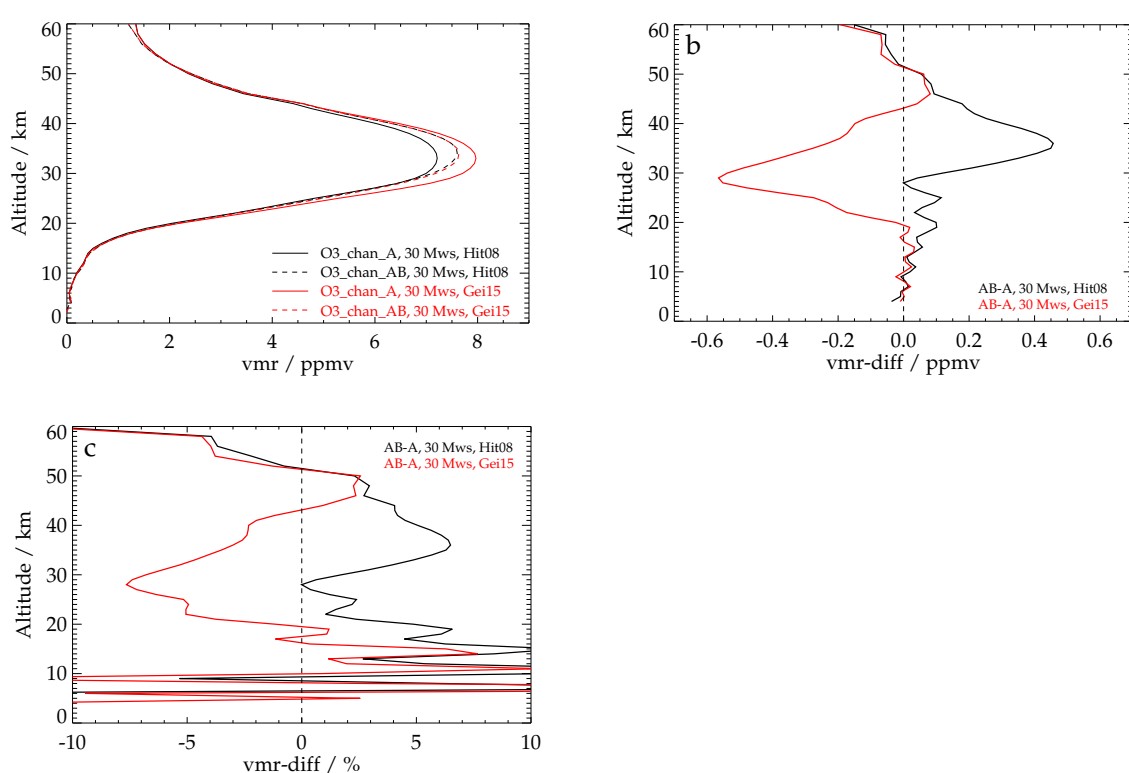

**Figure 6.** (a) Mean ozone profiles retrieved in MIPAS channels A (solid lines) and AB (dashed lines) using the HITRAN-2008 (black) and GEISA-2015 spectroscopy (red). Averaging was performed over 59 profiles from January 3, April 1, July 2 and October 2/3, 2009. (b) Absolute and (c) relative differences between the profiles retrieved in channels AB and A using HITRAN-2008 (black) and GEISA-2015 (red).

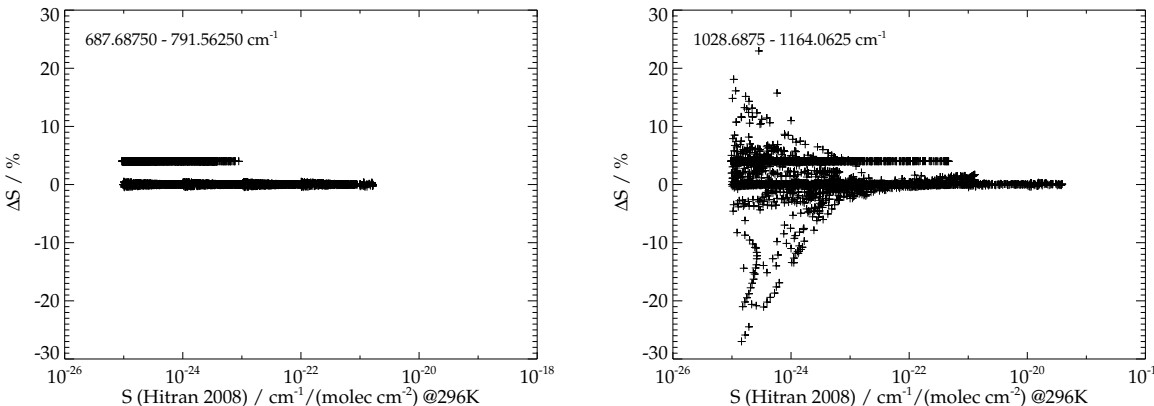

**Figure 7.** Relative differences ΔS between ozone line intensities of GEISA-2015 and HITRAN-2008 for the microwindows in MIPAS channel A (left) and AB (right).

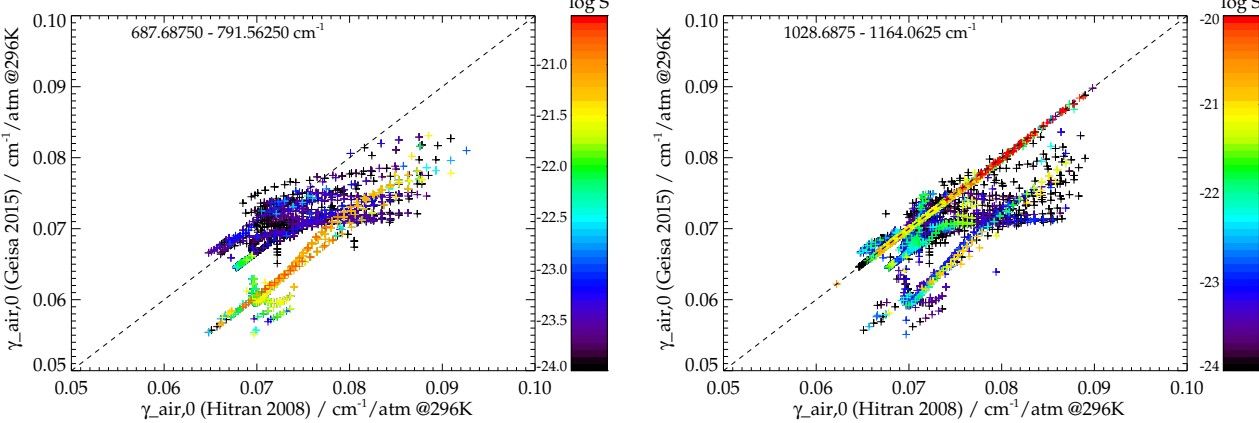

**Figure 8.** Left: Air-broadened halfwidths $\gamma_{air,0}$ of GEISA-2015 versus HITRAN-2008 for the microwindows in MIPAS channel A. Right: Same as left, but for the microwindows in channel AB. The halfwidths are colour-coded by the logarithm of the HITRAN-2008 line intensities.

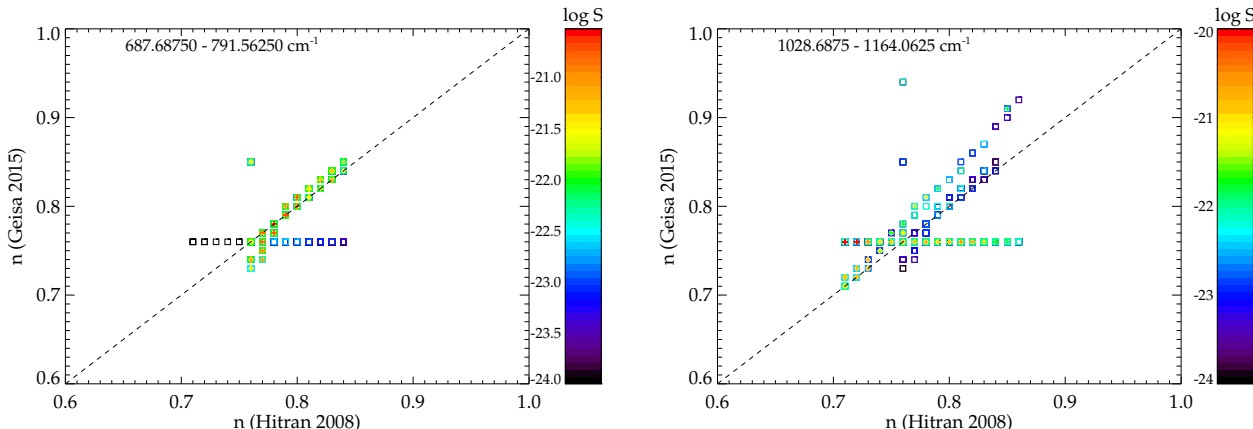

**Figure 9.** Left: Coefficients $n$ of temperature dependence of the air-broadened halfwidths of GEISA-2015 versus HITRAN-2008 for the microwindows in MIPAS channel A. Right: Same as left, but for the microwindows in channel AB. The coefficients are colour-coded by the logarithm of the Hitran-2008 line intensities. Plotted squares/pluses denote coefficients associated to line strengths lower/larger than $1 \times 10^{-22}$ cm$^{-1}$ / molec cm$^{-2}$@296K.

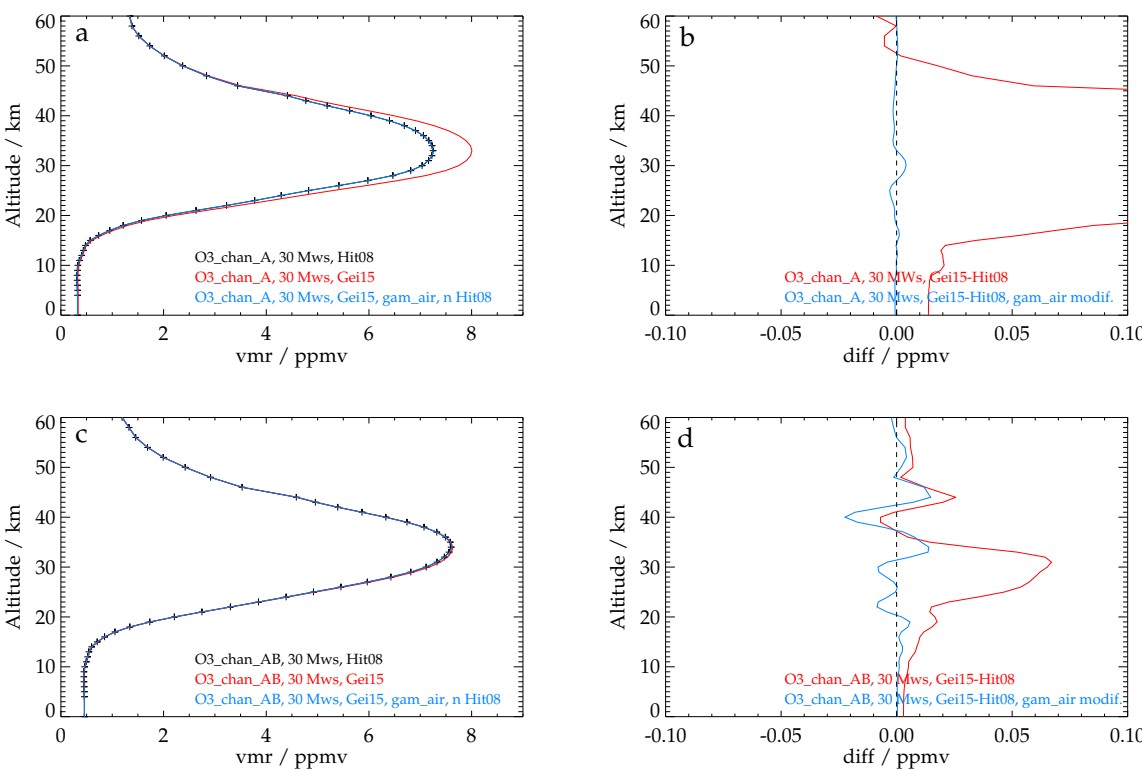

**Figure 10.** (a) Mean ozone profiles of 59 orbits from January 3, April 1, July 2 and October 2/3, 2009, retrieved in MIPAS channel A using the HITRAN-2008 (black), the GEISA-2015 (red) and a modified GEISA-2015 spectroscopy (blue). The modification consists in replacement of the air-broadened halfwidths and coefficients of temperature dependence by the HITRAN-2008-values. (b) Differences to the ozone profile resulting from use of the HITRAN-2008 spectroscopy. (c) Same as (a), but for MIPAS channel AB. (d) Differences to the ozone profile resulting from use of the HITRAN-2008 spectroscopy.

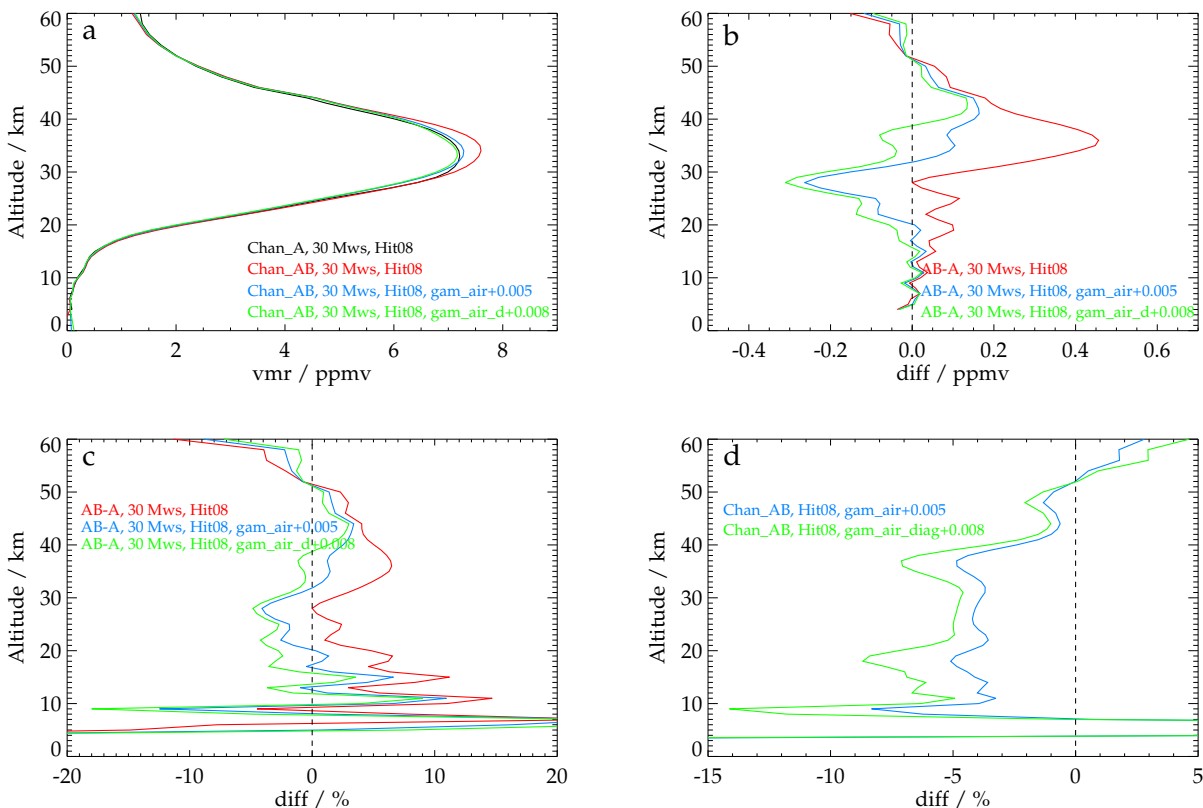

**Figure 11.** (a) Mean ozone profiles of 59 orbits from January 3, April 1, July 2 and October 2/3, 2009, retrieved in MIPAS channels A (black) and AB (red) using the HITRAN-2008 spectroscopy (cf. Figure 1). The blue profile results from retrieval in MIPAS channel AB after increase of the air-broadened halfwidths by a value of 0.005 cm$^{-1}$/atm@296K, and the green profile from retrieval after increase of the air-broadened halfwidths of the strong lines on the main diagonal in Figure 8 (right) by 0.008 cm$^{-1}$/atm@296K. (b) Absolute and (c) relative differences between retrieval in channels AB and A. (d) Differences between channel AB retrievals with increased air-broadened halfwidths and channel AB rertrievals using the unmodified HITRAN-2008 spectroscopy.

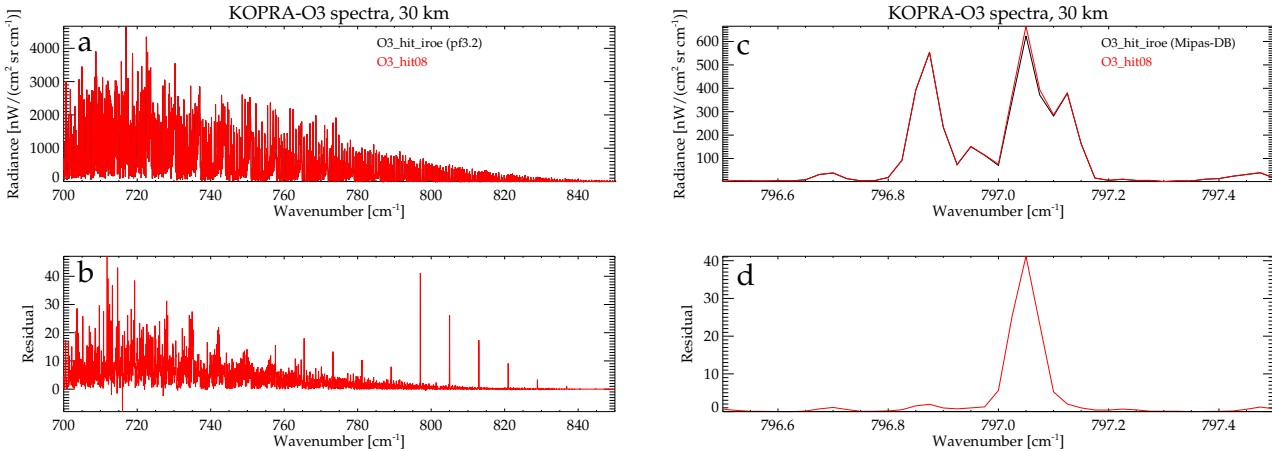

**Figure 12.** (a) Modelled ozone spectra for a tangent height of 30 km, using the MIPAS-pf3.2 (black) and HITRAN-2008 spectroscopy (red). (b) Difference between the two simulations. (c and d) Same as left, but for a zoom into the spectral region around 797.0 cm$^{-1}$.

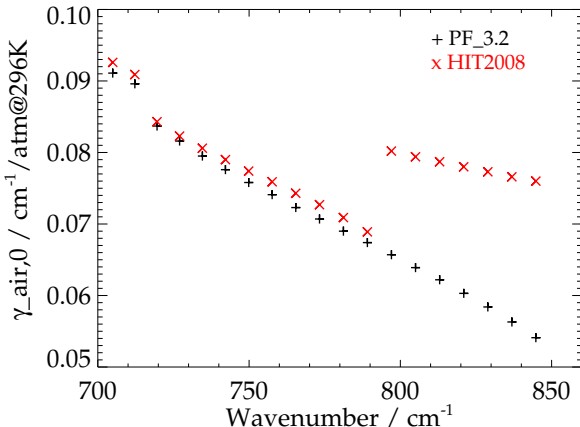

**Figure 13.** Air-broadened halfwidths $\gamma_{air,0}$ of several ozone lines of the fundamental $\nu_2$ ozone band in the MIPAS-pf3.2 (black) and HITRAN-2008 spectroscopy (red).

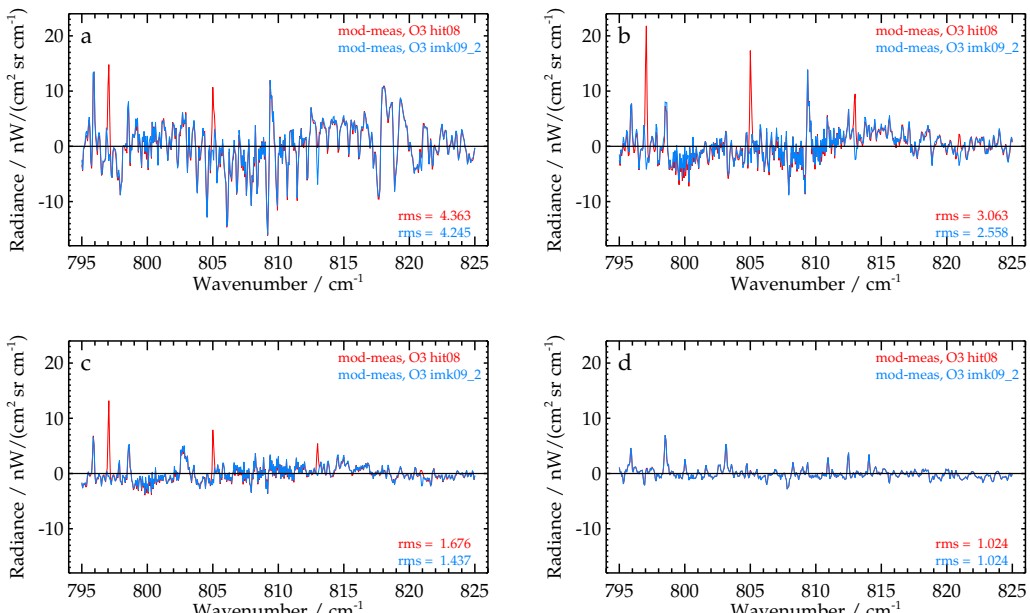

**Figure 14.** Residua between mean modelled and measured spectra using the HITRAN-2008 (red) and the MIPAS-pf3.2 ozone spectroscopy (blue) for tangent heights of (a) ∼14 km, (b) ∼20 km , (c) ∼30 km and (d) ∼44 km. Averaging was performed over 302 (14 km) and 442 spectra (higher altitudes), respectively, of 14 consecutive orbits (39680–39693) for the latitude range 30°S–30°N. The large red residua of the ozone lines at 797, 805 and 813 cm$^{-1}$ are caused by inappropriate air-broadening coefficients in the HITRAN-2008 spectroscopy.

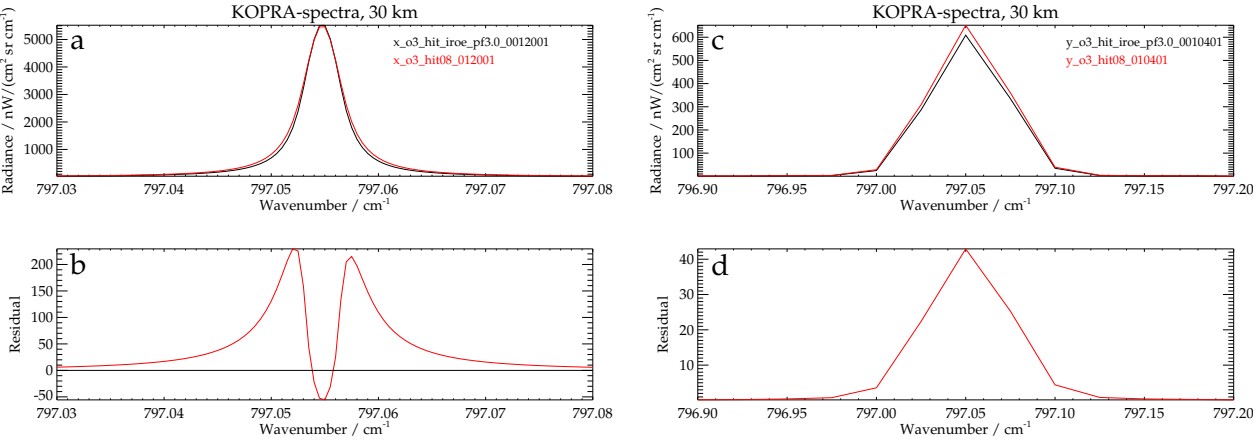

**Figure 15.** (a) Modelled spectral signature of the ozone line at 797.05 cm$^{-1}$ for a limb observation with a tangent height of 30 km using the MIPAS-pf3.2 (black) and the HITRAN-2008 spectroscopy (larger $\gamma_{air,0}$) (red), monochromatic lines. (b) Corresponding residual. (c and d) Same as left, but lines convolved with the apodized instrumental line shape.