# Peer review of "Differences in ozone retrieval in MIPAS channels A and AB: a spectroscopic issue"

_Atmospheric Measurement Techniques, 2018_

## Referee Comment (RC1) · Anonymous Referee #1 · 22 Apr 2018

GENERAL COMMENTS

The paper describes the tests performed to understand possible causes of the bias between ozone profiles retrieved from two different spectral bands of MIPAS measurements: channel A (685-970 cm-1) and channel AB (1020-1170 cm-1), mainly around the peak of the ozone profile.

Several possible sources of systematic errors are considered, but only spectroscopic errors seems responsible of the found differences. Tests with different spectroscopic databases (MIPAS pf3.2, HITRAN 1996, HITRAN 2004, HITRAN 2008, HITRAN 2016 and GEISA 2015) indicate that the major part of the channel AB-A differences might be caused by inconsistencies in air-broadened halfwidths of the lines of the databases. As a consequence, the authors suggest that the air-broadened halfwidths of ozone lines

in the spectral regions of MIPAS channel A as well as of channel should be reassessed both for the GEISA and for the HITRAN databases.

The paper is clear, and surely it is of interest for spectroscopists and people retrieving Ozone in the middle infrared. Therefore it can be published in AMT after minor revisions that are described below.

SPECIFIC COMMENTS:

Pag.4, Section 3. The description of the errors in MIPAS spectroscopic databases should be moved before Sect.7.1, where differences between several spectroscopic databases are quantified, and hence the estimation of the errors on line intensities and line width can be more useful.

Pag. 5, line 17: what 'completely different forward models' means ?

Pag. 5, lines 9-14: I think that these sentences may be misleading in the paper. Indeed, Laeng et al., 2014 shows that from the comparison between MIPAS Ozone with ACE-FTS and MLS, MIPAS is larger than both of them. Since O3 retrieved from channel AB is larger than O3 retrieved from channel A, we can deduce that the use of only spectral intervals in band A may reduce the differences with respect to ACE-FTS and MLS. However, we have to consider that positive differences between MIPAS and ACE-FTS are probably not due, or at least not only due, to spectroscopic issues, since ACE-FTS performes measurements in the same spectral regions as MIPAS and for the O3 retrieval mainly spectral points in the region of MIPAS band AB are used (see http://www.ace.uwaterloo.ca/misc/ACE-SOC-0027-ACE-FTS_Spectroscopy-version_3.5_Jan222016_Rev1A.pdf). Furthermore, the tests reported in this paper do not indicate which of the two bands A and AB has smallest spectroscopic errors, but only that there are inconsistencies between the two bands. Finally, the change of used spectral intervals in order to reduce the bias with other correlative measurements, that do not represent the true, may not always be correct.

[Figure]

MIPAS spectroscopic database pf 3.2 sometime is mentioned in the paper (e.g. Pag.9, line 4) as MIPAS spectroscopy, other times (e.g. Caption of Fig.5) as Mipas pf 3.0. Please use consistent terminology.

Last sentence of the paper: 'as far as ozone is concerned we recommend to use version pf3.2 of the MIPAS spectroscopy and not the latest update pf4.45, because the ozone data set in this compilation is identical with HITRAN-2008.' A reference to the spectroscopic database pf4.45 should be added. The presence of 'inappropriate halfwidths' in HITRAN 2008 and following versions seems to involve only the 790- 850 cm-1 spectral region.

TECHNICAL CORRECTIONS:

Pag..4, line 23: delete one 'the'

Pag.5, line 25: performed

Sect.5: Fig.4 is mentioned before Fig. 3

Pag. 6, line 15: by neglectING

Pag. 8: line 6: arE

Pag. 8: line 9: follOwing

In general, the figures with several plots are more readable if each plot is identified with a letter (a), b), c). . ..)

Fig.1 : x-label of bottom plot: please replace 'diff/ppmv' with 'diff/km'

---

## Referee Comment (RC2) · Anonymous Referee #2 · 30 Apr 2018

**General comments**

The present manuscript of Glatthor *et al.* provides an in-depth analysis of differences between ozone retrieval results from two separate MIPAS channels A (including ozone absorption lines from the $\nu_2$ and $\nu_3$ fundamentals) and AB (including lines of the $\nu_1$ and $\nu_3$ fundamentals). These differences are sizeable and significant and reach about 8 % at the maximum of the stratospheric ozone peak. After having ruled out model, retrieval and instrumental errors being responsible for this discrepancy, the authors compare retrieval results using different spectroscopic databases. They find that pressure broadening parameters in the various data sets are quite different and lead to the large discrepancy rather than inconsistencies in the line strength data.

[Figure]

The manuscript is well written and structured. The method is valid and figures were selected appropriately. The paper should therefore be published after the necessary corrections have been made.

The manuscript could possibly gain more widespread interest by i) including a comparison of pressure broadening parameters to including the MIPAS data base and ii) by quantifying the impact on total ozone columns. This would allow to better estimate the impact of this particular parameter on the existing bias between UV and IR comparison measurements (see Orphal et al., 2016, and references therein).

**Specific comments**

There are two possibly important omissions in the paper. As already pointed out above, the MIPAS database/spectroscopy deserves a short presentation so that similarities and differences with respect to the other data bases become clear. It would also be helpful to see a detailed comparison of line-broadening parameters between MIPAS and HITRAN and MIPAS and GEISA (such as in Figs. 8 and 9 for HITRAN and GEISA) to better understand differences in the data bases. This also because MIPAS is finally recommended to be preferred over the other data sets. The other issue is that line intensities (see line strengths of $\nu_2 = 1 \leftarrow 0$, $(J+1, J+1, 1) \leftarrow (J, J, 0)$ transitions in Fig. 13, for example) are compared using reference temperatures at room, but at stratospheric temperatures the lower state energies (and to a lesser extent partition sums) also contribute. The quoted line strength uncertainty might thus be too optimistic. While partition sums cannot lead to an inter-band bias, lower state energies can. For the sake of completeness a discussion of the impact of possible differences in lower state energies or a comparison of low temperature intensities would be required.

The manuscript preparation guidelines request that "works cited in a manuscript

should be accepted for publication or published already" and the authors should therefore avoid utilizing personal communications. The communications used are not really required and seem to be problematic. For example, in Section 3 (*Error estimates of ozone lines and band intensities*), a pers. communication (J.–M. Flaud) is given to motivate relative errors of the three fundamentals. Eq. (1) indicates that the relative error is the same for the $\nu_1$, $\nu_2$ and $\nu_3$ bands. However, the comparison of experimental data with intensity calculations from the same author shows that the agreement in the $\nu_2$ cold band is usually worse than in the other two fundamental bands (See section 5.2.2. of Wagner et al., 2002). This information therefore seems to be conflicting. Later it is stated that "These inappropriate halfwidths (M. Birk, pers. comm.) are the reason for the stronger ozone lines in the model spectrum using HITRAN-2008 data in Figure 12. This deficiency is still present in later versions up to HITRAN-2016." A priori, it is not clear which set of half widths should be correct and which not and why these half widths cause problems. Non-continuous behaviour is visible in both data sets (see Fig. 13 right). Wouldn't it be more informative and decisive to show the direct comparison between modelled and experimental spectra ?

The study of Janssen et al. (2016) needs to be mentioned in the paper. It has evident methodological links and has already identified differences in pressure broadening parameters between GEISA (version of 2011) and HITRAN (version of 2012) being the main reason for ozone column retrieval differences in the $\nu_3$ spectral region at 10 μm. It seems that the surprising effect (section 8: *Additional observations*) of systematic biases in the air broadened half width potentially leading to positive and negative feed-backs depending on the optical thickness of the atmosphere is discussed there as well.

Fig. 6 requires correction. On the one hand some technical information on averaging kernel thresholds and orbit numbers are probably not very informative. On the other hand, the difference plot and the absolute values of the GEISA retrievals are not

compatible in the altitude range $< 10$ km. There is a clear offset (AB $-$ B $> 0$) between the two bands on the left panel, but the difference plot on the right shows AB $=$ B.

Absolute deviations at the per cent level are difficult to perceive on the logarithmic scale. The left plot of Fig. 13 should show the relative deviation between intensities from HIT-08 and PF-3.0.

**Technical corrections**

- p. 1, l. 8 : Thus spectroscopic ... $\rightarrow$ Thus, spectroscopic ...

- p. 3, l. 27–29 : Phrase is incomplete/wrong

- p. 3, l. 29 : schema $\rightarrow$ scheme

- p. 4, l. 22 : The acronym IAA appears for the first time. Please explain.

- p. 7, l. 18 : .Since $\rightarrow$ . Since

- p. 7, l. 22 : basing $\rightarrow$ based (?)

- p. 8, l. 6 : ar $\rightarrow$ are

- p. 10, l. 4–5 : To check, if ... $\rightarrow$ We performed two additional tests to check if ...

- p. 11, l. 6 : could be widely excluded $\rightarrow$ could be excluded

- p. 11, l. 7-9 : Repeated use of also. Delete one instance.

- p. 22, Fig. 7 : The top panels might be omitted. They do not provide information that is not already contained in the bottom panels. The legend would need to be adjusted accordingly.

- p. 23, Fig. 9 caption : Add units to line strength values.

- p. 26, Fig. 13 : Units are missing on vertical axes.

**References**

Orphal J., Staehelin J., Tamminen J., Braathen G., De Backer M. R., Bais A. F., Balis D., Barbe A., Bhartia P. K., Birk M., Burkholder J. B., Chance K. V., von Clarman T., Cox A., Degenstein D., Evans R., Flaud J. M., Flittner D., Godin-Beckmann S., Gorshelev V., Hare E., Janssen C., Kyrölä E., Mcelroy T., McPeters R., Pastel M., Petersen M., Petropavlovskikh I., Picquet-Varrault B., Pitts M., Labow G., Rotger-Langerau M., Leblanc T., Lerot C., Liu X., Moussay P., Redondas A., Van Roozendael M., Sander S. P., Schneider M., Serdyuchenko A., Veefkind P., Viallon J., Viatte C., Wagner G., Weber M., Wielgosz R. I. and Zahner C.: Absorption cross-sections of ozone in the ultraviolet and visible spectral regions: Status report 2015, *J. Mol. Spectrosc.* **327**, 105–121, 2016.

Wagner, G., Birk, M., Schreier, F. and Flaud, J.–M.: Spectroscopic database for ozone in the fundamental spectral regions, *J. Geophys. Res.,* **107**, 4626, 2002.

Janssen, C., Boursier, C., Jeseck, P., and Té, Y.: Line parameter study of ozone at 5 and 10 μm using atmospheric FTIR spectra from the ground: A spectroscopic database and wavelength region comparison, *J. Mol. Spectrosc.* **326**, 48–59, 2016.
* * *

---

## Author Comment (AC1) · 7 Jun 2018

We thank reviewer 1 for her/his helpful comments. Please find below our responses describing how the manuscript has been modified with respect to the comments. Blue passages denote the changes or updates in the revised manuscript.

**Specific Comments**

Comment: *"Pag.4, Section 3. The description of the errors in MIPAS spectroscopic databases should be moved before Sect.7.1, where differences between several spectroscopic databases are quantified, and hence the estimation of the errors on line*

[Figure]

*intensities and line width can be more useful."*

Reply: Similar to the description of the MIPAS experiment and of the retrieval setup in Section 2, the error estimates for ozone lines in the MIPAS and HITRAN spectroscopic databases are a prerequisite for our investigations. Therefore we find it more appropriate to leave the description of these errors in Section 3 prior to the retrieval section. To address the reviewer's point, we will compare the spectroscopic errors with the VMR differences in Sections 4 and 7.1 by adding the sentences "This difference is larger than the relative error in line intensity given in Eqs. 1 and 2 for the strongest and medium scale ozone lines (at least for transitions with low to medium-sized rotational quanta JU and KU)." at page 4, line 26, and "Consequently, these differences are also larger than the relative errors in line intensity given in Eqs. 1 and 2." at page 7, line 15.

Comment: *"Pag. 5, line 17: what 'completely different forward models' means ?"*

Reply: This phrase is maybe a bit incomprehensible, because the forward model KOPRA we use at IMK/IAA has not been introduced before. For this reason we will replace the first sentence of the second paragraph of Section 2 by "To reinvestigate the channel AB-A bias in retrieved ozone, retrievals using the the processor of the Institut für Meteorologie und Klimaforschung and the Instituto de Astrofísica de Andalucía (IMK/IAA) were performed for 59 MIPAS orbits from January 3, April 1, July 2 and October 2-3, 2009. This processor uses the Karlsruhe Optimized and Precise Radiative Algorithm (KOPRA) (Stiller, 2000) for radiative transfer calculations and the Retrieval Control Program (RCP) of IMK/IAA for inverse modelling of spectra. ". Further we will change the critisised phrase into "a different radiative transfer model". Because by these modifications the acronym KOPRA is already explained in Section 2, we will change the subsequent sentence (page 5, lines 19/20) into "This agreement widely excludes the hypothesis that the bias is caused by deficiencies in the KOPRA

forward model used at IMK."

Comment: *"Pag. 5, lines 9-14: I think that these sentences may be misleading in the paper. Indeed, Laeng et al., 2014 shows that from the comparison between MIPAS Ozone with ACE-FTS and MLS, MIPAS is larger than both of them. Since O3 retrieved from channel AB is larger than O3 retrieved from channel A, we can deduce that the use of only spectral intervals in band A may reduce the differences with respect to ACE-FTS and MLS. However, we have to consider that positive differences between MIPAS and ACE-FTS are probably not due, or at least not only due, to spectroscopic issues, since ACE-FTS performes measurements in the same spectral regions as MIPAS and for the O3 retrieval mainly spectral points in the region of MIPAS band AB are used (see http://www.ace.uwaterloo.ca/misc/ACE-SOC-0027-ACE-FTS_Spectroscopy-version_3.5_Jan222016_Rev1A.pdf). Furthermore, the tests reported in this paper do not indicate which of the two bands A and AB has smallest spectroscopic errors, but only that there are inconsistencies between the two bands. Finally, the change of used spectral intervals in order to reduce the bias with other correlative measurements, that do not represent the true, may not always be correct."*

Reply: We do not quite understand the referee's arguments in this comment. First of all, Laeng et al. (2014, Fig. 5) indeed show that the MIPAS ozone VMRs are larger than those of MLS at nearly all altitudes, but there is no general positive bias with respect to ACE-FTS. MIPAS ozone VMRs are up to 3% larger than those of ACE-FTS below 30 km, but up to 2% lower between 30 and 45 km. Between 45 and 55 km MIPAS ozone is even more than 10% lower than ACE-FTS ozone. Secondly, ACE-FTS does not perform measurements (ozone retrievals) in the same spectral region as MIPAS. ACE-FTS uses the spectral region 1027–1059 $cm^{-1}$ (see document cited above), but MIPAS (data version O3_V5R_224) the region 687–791 $cm^{-1}$. Only above 50 km two channel AB microwindows at 1029–1031 and 1038–1039 $cm^{-1}$ are added.

[Figure]

Thus, differences between ACE-FTS and MIPAS can well have spectroscopic causes. We agree with the referee's statement that "the tests reported in this paper do not indicate which of the two bands A and AB has smallest spectroscopic errors". Just as he/she concludes, we only want to show "that there are inconsistencies between the two bands." Further, the referee might be right by stating that "the change of used spectral intervals ... may not always be correct". But there is justification for such a change, if a similar bias to several correlative instruments can be reduced in doing so. With this we can at least provide an explanation of the discrepancies encountered.

Comment: *"MIPAS spectroscopic database pf 3.2 sometime is mentioned in the paper (e.g. Pag.9, line 4) as MIPAS spectroscopy, other times (e.g. Caption of Fig.5) as Mipas pf 3.0. Please use consistent terminology. "*

Reply: We agree and will speak of MIPAS pf3.2 throughout the updated manuscript. We were a bit unprecise, because the ozone spectroscopy in MIPAS pf3.2 is the same as in MIPAS pf3.0.

Comment: *"Last sentence of the paper: 'as far as ozone is concerned we recommend to use version pf3.2 of the MIPAS spectroscopy and not the latest update pf4.45, because the ozone data set in this compilation is identical with HITRAN-2008.' A reference to the spectroscopic database pf4.45 should be added. The presence of 'inappropriate halfwidths' in HITRAN 2008 and following versions seems to involve only the 790- 850 cm-1 spectral region."*

Reply: We will add the reference "Flaud, J.-M., Perrin, A., and Ridolfi, M.: New release of the MIPAS spectroscopic database: hitran_mipas_pf_v4.45, Presentation at MIPAS QWG 38, ESA-ESRIN, 18-19 February 2015." for the spectroscopic database pf4.45.

[Figure]

Concerning inappropriate halfwidths in HITRAN-2008: We showed one example of an obviously unphysical step in halfwidths at 797.05 cm$^{-1}$ in HITRAN-2008 and subsequent editions. However, we can not draw general conclusions about the spectral ranges of inappropriate halfwidths in the HITRAN data bases. This issue has to be left to spectroscopists.

**Technical Corrections**

The requested technical corrections will be performed. Fig. 4 will be interchanged with Fig. 3 to obtain a consecutive discussion of the figures. In figures with several plots each plot will be identified with a letter. Finally, "diff / ppmv" will be replaced by "diff / km" in Fig. 1.

**References**

Laeng, A., et al.: Validation of MIPAS IMK/IAA V5R_O3_224 ozone profiles, Atmos. Meas. Tech., 7, 3971–3987, www.atmos-meas-tech.net/7/3971/2014/, 2014.

---

## Author Comment (AC2) · 7 Jun 2018

Many thanks for reading our manuscript and your helpful comments. Please find below our responses describing how the manuscript has been modified with respect to your annotations. Blue passages denote changes or updates in the revised manuscript.

**General Comments**

For clarification: We do not use lines of the fundamental $\nu_3$ band for channel A retrievals, but lines of the $\nu_2$ band and higher transitions in the spectral range 687–791 cm$^{-1}$ only.

[Figure]

Comment: *"The manuscript could possibly gain more widespread interest by i) including a comparison of pressure broadening parameters to including the MIPAS data base and ii) by quantifying the impact on total ozone columns. This would allow to better estimate the impact of this particular parameter on the existing bias between UV and IR comparison measurements (see Orphal et al., 2016, and references therein)."*

Reply: (i) We think that inclusion of the MIPAS pf3.2 database in the comparison of pressure broadening parameters in the main manuscript is not very conductive, because we do not want to find the reasons for the relatively small differences between ozone retrievals using the MIPAS pf3.2 and the HITRAN-2008 spectroscopy. To address the referee's suggestion, we will add the following paragraph at the end of Section 7.1: "The rather good agreement between the channel A as well as the channel AB retrievals using the MIPAS spectroscopy or the HITRAN 2004 edition and later ones indicates largely consistent spectroscopic parameters of identical ozone lines in these databases for the spectral range of the channel A and AB microwindows. Therefore a comparison between the line parameters of the MIPAS and HITRAN databases is not presented here, but as supplemental material only." and show the requested comparison as supplemental material. To emphasize the good intra-band agreement between most of the channel A and AB profiles (MIPAS vs HITRAN) we will add two graphs showing the absolute channel A and AB profiles to Figure 5. Further, we will slightly change the first paragraph of Section 7.2.2 into "The retrieval results also indicate mostly consistent spectral parameters in HITRAN-2008 and GEISA-2015 for the ozone lines used in MIPAS channel AB, but considerable spectroscopic differences in the region of the channel A microwindows. In the following, we will compare the HITRAN-2008 and GEISA-2015 ozone lines applied in channel A as well as in channel AB to identify the parameters responsible for these differences.

(ii) To consider the referee's second point, we will add a fourth graph to Figure 11

showing the relative differences between channel AB retrievals using the unchanged pressure broadening coefficients and the two sets of modified coefficients. Since these differences are around -4% and -5% over a large altitude range, they are a good estimate for the respective changes in ozone columns. We will shortly discuss this finding in Section 7.3 in comparison with the results of Schneider et al. (2008), who found a systematic difference of 4–5 % between IR measurements in the spectral range 991–1007 cm$^{-1}$ and UV observations. Our results show that, beside the re-scaling of line strengths of the $\nu_3$ and $\nu_1$ lines by 4% as discussed in Smith et al. (2012), a change of air broadening coefficients can lead to a similar adjustment to the UV measurements.

**Specific Comments**

Comment: *"There are two possibly important omissions in the paper. As already pointed out above, the MIPAS database/spectroscopy deserves a short presentation so that similarities and differences with respect to the other data bases become clear. It would also be helpful to see a detailed comparison of line-broadening parameters between MIPAS and HITRAN and MIPAS and GEISA (such as in Figs. 8 and 9 for HITRAN and GEISA) to better understand differences in the data bases. This also because MIPAS is finally recommended to be preferred over the other data sets. The other issue is that line intensities (see line strengths of 2 = 1 0, (J + 1, J + 1, 1) (J, J, 0) transitions in Fig. 13, for example) are compared using reference temperatures at room, but at stratospheric temperatures the lower state energies (and to a lesser extent partition sums) also contribute. The quoted line strength uncertainty might thus be too optimistic. While partition sums cannot lead to an inter-band bias, lower state energies can. For the sake of completeness a discussion of the impact of possible differences in lower state energies or a comparison of low temperature intensities would be required. "*
Reply: A short presentation of the MIPAS spectroscopy is already given in the Introduction on page 2, lines 2–4. We will add some more information on the MIPAS spectroscopy here. As mentioned above, we will present and discuss a comparison between the spectral parameters in MIPAS pf3.2 and in HITRAN-2008 such as in Figs. 8 and 9 as supplemental material. As requested by the referee, we also performed a comparison of the lower state energies of the corresponding lines in MIPAS pf3.2, HITRAN-2008 and GEISA-2015 for the spectral region of our microwindows. Except of some very weak lines there is perfect agreement between the lower state energies. We will discuss the potential bias due to inconsistent lower state energies and the result of our inspection at the end of the first paragraph on page 9.

Comment: *" The manuscript preparation guidelines request that "works cited in a manuscript should be accepted for publication or published already" and the authors should therefore avoid utilizing personal communications. The communications used are not really required and seem to be problematic. For example, in Section 3 (Error estimates of ozone lines and band intensities), a pers. communication (J.-M. Flaud) is given to motivate relative errors of the three fundamentals. Eq. (1) indicates that the relative error is the same for the 1, 2 and 3 bands. However, the comparison of experimental data with intensity calculations from the same author shows that the agreement in the 2 cold band is usually worse than in the other two fundamental bands (See section 5.2.2. of Wagner et al., 2002). This information therefore seems to be conflicting. Later it is stated that "These inappropriate halfwidths (M. Birk, pers. comm.) are the reason for the stronger ozone lines in the model spectrum using HITRAN-2008 data in Figure 12. This deficiency is still present in later versions up to HITRAN-2016." A priori, it is not clear which set of half widths should be correct and which not and why these half widths cause problems. Non-continuous behaviour is visible in both data sets (see Fig. 13 right). Wouldn't it be more informative and*
*decisive to show the direct comparison between modelled and experimental spectra?
"*

Reply: To address the referee's objection against personal communications we will change the phrase on page 2, lines 26-28 into "Since the uncertainties in line intensity of many lines of the $\nu_2$ and $\nu_1/\nu_3$ fundamentals observed in channels A and AB, respectively, have been determined to be less than 2% (Wagner et al., 2002) ...". Further we will remove the phrase "(M. Birk, pers. comm.)" on page 10, line 20. However, the error estimates given in Section 3 are from an internal technical note by J.-M. Flaud and C. Piccolo for MIPAS data evaluation only and can not be cited in a more convenient way. The referee critizises that the relative error given in Eq. 1 "is the same for the 1, 2 and 3 bands". But exactly these error estimates were provided by Flaud and Piccolo. Wagner et al. (2002) with Flaud as co-author indeed state a lower accuracy for the $\nu_2$ band at the end of Section 5.2.2, but obviously a weak degradation only ("The results are a little bit worse for the $\nu_2$ band ..."). Coming to the issue of inappropriate halfwidths: As correctly noticed by the referee, non-continuous behaviour is visible in both data sets (Fig. 13, right), but the jump in HITRAN-2008 at 797.05 cm$^{-1}$ is considerably larger than the jump in MIPAS pf3.2 (and HITRAN-2008) at 713 cm$^{-1}$. As suggested, we will add the results of broadband retrievals in the region 795–825 cm$^{-1}$, which clearly show that the halfwidths of MIPAS pf3.2 of the respective lines at 797.05, 805.02 and 812.99 cm$^{-1}$ lead to much better agreement with the measurements than those of HITRAN-2008.

Comment: *" The study of Janssen et al. (2016) needs to be mentioned in the paper. It has evident methodological links and has already identified differences in pressure broadening parameters between GEISA (version of 2011) and HITRAN (version of 2012) being the main reason for ozone column retrieval differences in the 3 spectral region at 10 $\mu$m. It seems that the surprising effect (section 8: Additional observations)*

*of systematic biases in the air broadened half width potentially leading to positive and negative feedbacks depending on the optical thickness of the atmosphere is discussed there as well. "*

Reply: After having read the Janssen et al. (2016) paper, we think that its main link to our paper is the discussion in Section 3.2.2 (Sensitivity on pressure broadening coefficient). In this section these authors discuss the results of Table 4 and show the "striking feature" that for lines of the $\nu_3$ band an increase in $\gamma\_air$ similar as an increase in line intensity leads to a negative change in the ozone column. This is consistent to the results in Section 8 of our manuscript. We will cite Janssen et al. (2016) and mention their similar findings after the last sentence of Section 8 (page 10) in our manuscript.

Comment: *" Fig. 6 requires correction. On the one hand some technical information on averaging kernel thresholds and orbit numbers are probably not very informative. On the other hand, the difference plot and the absolute values of the GEISA retrievals are not compatible in the altitude range < 10 km. There is a clear offset (AB - B > 0) between the two bands on the left panel, but the difference plot on the right shows AB = B. "*

Reply: The referee is right. The inconsistency below 10 km occured, because the cloud filter was switched on for calculation of the mean differences, but erroneously not applied for calculation of the mean absolute profiles. This error will be corrected. Further, we will remove technical information on averaging kernels, orbit numbers etc. in Fig. 6 and in similar figures.

Comment: *" Absolute deviations at the per cent level are difficult to perceive on the*

*logarithmic scale. The left plot of Fig. 13 should show the relative deviation between intensities from HIT-08 and PF-3.0. "*

Reply: We agree and will change Fig. 13 accordingly.

**Technical corrections**

Comment: *"p. 3, l. 27–29 : Phrase is incomplete/wrong"*

Reply: We can not find a clear omission or error in these sentences and ask the referee to give a more specific comment, please.

Comment: *"p. 4, l. 22 : The acronym IAA appears for the first time. Please explain.*

Reply: Since the acronym IAA will already be explained in Section 2 of the updated manuscript (cf. reply to referee 1), it does no longer need to be explained here.

All other technical corrections will be performed as suggested.

**References**

Schneider, M., Redondas, A., Hase, F., Guirado, C., Blumenstock, T., and Cuevas, E.: Comparison of ground-based Brewer and FTIR total column $O_3$ monitoring techniques, Atmos. Chem. Phys., 8, www.atmos-chem-phys.net/8/5535/2008/, 5535–5550, 2008.